# Investigating the replicability of preclinical cancer biology

**Timothy M Errington[1]\*, Maya Mathur[2], Courtney K Soderberg[1], Alexandria Denis[1†], Nicole Perfito[1‡], Elizabeth Iorns[3], Brian A Nosek[1,4]**

[1]Center for Open Science, Charlottesville, United States; [2]Quantitative Sciences Unit, Stanford University, Stanford, United States; [3]Science Exchange, Palo Alto, United States; [4]University of Virginia, Charlottesville, United States

**Abstract** Replicability is an important feature of scientific research, but aspects of contemporary research culture, such as an emphasis on novelty, can make replicability seem less important than it should be. The Reproducibility Project: Cancer Biology was set up to provide evidence about the replicability of preclinical research in cancer biology by repeating selected experiments from high-impact papers. A total of 50 experiments from 23 papers were repeated, generating data about the replicability of a total of 158 effects. Most of the original effects were positive effects (136), with the rest being null effects (22). A majority of the original effect sizes were reported as numerical values (117), with the rest being reported as representative images (41). We employed seven methods to assess replicability, and some of these methods were not suitable for all the effects in our sample. One method compared effect sizes: for positive effects, the median effect size in the replications was 85% smaller than the median effect size in the original experiments, and 92% of replication effect sizes were smaller than the original. The other methods were binary – the replication was either a success or a failure – and five of these methods could be used to assess both positive and null effects when effect sizes were reported as numerical values. For positive effects, 40% of replications (39/97) succeeded according to three or more of these five methods, and for null effects 80% of replications (12/15) were successful on this basis; combining positive and null effects, the success rate was 46% (51/112). A successful replication does not definitively confirm an original finding or its theoretical interpretation. Equally, a failure to replicate does not disconfirm a finding, but it does suggest that additional investigation is needed to establish its reliability.

**\*For correspondence:**
tim@cos.io

**Present address:** †Fordham University School of Law, New York, United States; ‡Rarebase, Palo Alto, United States

## Introduction

Science is a system for accumulating knowledge. Independent researchers and teams study topics and make claims about nature based on the evidence that they gather. They also share their work so that others can evaluate, extend, or challenge the evidence and claims. The accumulation of evidence weeds out claims that are unreliable, and supports the development of new models and theories. Over time, uncertainty declines and the accumulated knowledge provides useful descriptions, effective predictions, and a better understanding of nature. Humanity applies this system in service of creating knowledge, treating disease, and advancing society.

An important feature of this system is replication (*Hempel, 1968*; *Musgrave, 1970*; *Nosek and Errington, 2020a*; *Salmon and Glymour, 1999*). A scientific claim is said to be replicable if it is supported by new data. However, it is often not straightforward to decide if a claim is supported by new data or not. Moreover, the success or failure of an attempt to replicate rarely provides a definitive answer about the credibility of an original claim. When the replication attempt is successful, confidence in the reliability of the claim increases, but that does not mean that the claim is valid: a finding can be both replicable and invalid at the same time. Repeated successful replications can help to eliminate alternative explanations and potential confounding influences, and therefore increase

confidence in both reliability and validity, but they might not eliminate all confounding influences. It is possible that the original experiment and all the replication attempts could be invalidated by a common shortcoming in experimental design.

When a replication attempt is not successful, it is possible that the original was a false positive – noise mistaken as a signal. It is possible that the original claim was overly generalized and is only replicable under a much narrower range of conditions than was originally believed. It is also possible that the methodology necessary to produce the evidence is not sufficiently defined or understood, or that the theoretical explanation for why the finding occurred is incorrect. Failures in implementing experimental protocols may also result in replication attempts being uninformative.

All of these possibilities are ordinary and can occur when researchers have been rigorous in their work. What should not be ordinary is persistent failure to recognize that some scientific claims are not replicable. Science advances via self-correction, the progressive identification and elimination of error. Self-correction requires a healthy verification process that recognizes non-replicability, eliminates unproductive paths, and redirects attention and resources to promising directions. A failure to recognize that some claims are not replicable can foster overconfidence, underestimate uncertainty, and hinder scientific progress.

There is accumulating evidence that non-replicability may be occurring at higher rates than recognized, potentially undermining credibility and self-correction. Theoretical analyses point to a system of incentives that prioritizes innovation over verification, leading to infrequent efforts to replicate findings and to behaviors that could reduce the replicability of published findings such as selective reporting, presenting exploratory findings as confirmatory tests, and failures of documentation, transparency, and sharing (*Casadevall and Fang, 2012*; *Gelman and Loken, 2013*; *Greenwald, 1975*; *Kimmelman et al., 2014*; *Makel et al., 2012*; *Nosek et al., 2012*; *Rosenthal, 1979*). For instance, one theoretical analysis estimated that more than half of research findings are false (*Ioannidis, 2005*). And in a survey, 60% of biologists who responded reported that they had failed to replicate their own results, and more than 75% had failed to replicate results from a different lab (*Baker, 2016*).

Large-scale replication studies in the social and behavioral sciences provide evidence of replicability challenges (*Camerer et al., 2016*; *Camerer et al., 2018*; *Ebersole et al., 2016*; *Ebersole et al., 2020*; *Klein et al., 2014*; *Klein et al., 2018*; *Open Science Collaboration, 2015*). In psychology, across 307 systematic replications and multisite replications, 64% reported statistically significant evidence in the same direction and effect sizes 68% as large as the original experiments (*Nosek et al., 2021*).

In the biomedical sciences, the ALS Therapy Development Institute observed no effectiveness of more than 100 potential drugs in a mouse model in which prior research reported effectiveness in slowing down disease, and eight of those compounds were tried and failed in clinical trials costing millions and involving thousands of participants (*Perrin, 2014*). Of 12 replications of preclinical spinal cord injury research in the FORE-SCI program, only two clearly replicated the original findings – one under constrained conditions of the injury and the other much more weakly than the original (*Steward et al., 2012*). And, in cancer biology and related fields, two drug companies (Bayer and Amgen) reported failures to replicate findings from promising studies that could have led to new therapies (*Prinz et al., 2011*; *Begley and Ellis, 2012*). Their success rates (25% for the Bayer report, and 11% for the Amgen report) provided disquieting initial evidence that preclinical research may be much less replicable than recognized. Unfortunately, because of proprietary concerns, very little information was made available on the studies that failed to replicate, on the replication methodology, or on the particular barriers encountered for replicating the findings. This lack of transparency makes it difficult to ascertain the reasons for failures to replicate and critique the basis of the claims.

In the Reproducibility Project: Cancer Biology, we sought to acquire evidence about the replicability of preclinical research in cancer biology by repeating selected experiments from 53 high-impact papers published in 2010, 2011, and 2012 (*Errington et al., 2014*). We describe in a companion paper (*Errington et al., 2021b*) the challenges we encountered while repeating these experiments. These barriers include: shortcomings in documentation of the original methodology; failures of transparency in original findings and protocols; failures to share original data, reagents, and other materials; methodological challenges encountered during the execution of the replication experiments. These challenges meant that we only completed 50 of the 193 experiments (26%) we planned to repeat. The 50 experiments that we were able to complete included a total of 158 effects that could be compared with the same effects in the original paper. It was common for experiments to have multiple

effects, such as assessing whether an intervention affected both tumor burden and overall survival, or assessing the impact that depleting different genes has on cellular proliferation.

In this paper, we report the results of a meta-analysis of all these comparisons. There is no single method for assessing the success or failure of replication attempts (*Mathur and VanderWeele, 2019*; *Open Science Collaboration, 2015*; *Valentine et al., 2011*), so we used seven different methods to compare the effect reported in the original paper and the effect observed in the replication attempt (see Results). Six of these methods were dichotomous (i.e., replication success/failure) and one was not.

In total, 136 of the 158 effects (86%) reported in the original papers were positive effects – the original authors interpreted their data as showing that a relationship between variables existed or that an intervention had an impact on the biological system being studied. The other 22 (14%) were null effects – the original authors interpreted their data as not showing evidence for a meaningful relationship or impact of an intervention. Furthermore, 117 of the effects reported in the original papers (74%) were supported by a numerical result (such as graphs of quantified data or statistical tests), and 41 (26%) were supported by a representative image or similar. For effects where the original paper reported a numerical result for a positive effect, it was possible to use all seven methods of comparison. However, for cases where the original paper relied on a representative image (without a numerical result) as evidence for a positive effect, or when the original paper reported a null effect, it was not possible to use all seven methods.

## Results

In this section we discuss the seven different methods that we used to assess replication attempts, and report what we found when we used these methods to compare the effects reported in the original papers and the effects observed in the replications. The results are reported in *Table 1*. We display the results of original positive effects and original null effects separately; we also display cases where the original effect was reported as a numerical value separate from cases where the original effect was reported as a representative image. In some cases we conducted two or more internal replication experiments for the same original effect, which increased the total number of outcomes from 158 to 188 (see Materials and methods). In the text of this article we mostly report and discuss our results in terms of effects, the relevant tables and figures report the results by outcome, effect, experiment, and paper.

The nested structure of outcomes within effects, effects within experiments, and experiments within papers provides different ways to characterize the results, and it is possible for some effects within an experiment to replicate successfully while other effects in the same experiment fail to replicate. However, the results are similar irrespective of whether we look at them by paper (23 in total), by experiment (50 in total), by effect (158 in total), or by outcome (188 in total). We also use a number of strategies for aggregating data across effects and experiments, but observe very similar findings regardless of method used for aggregation (see Tables S1–S3 in *Supplementary file 1*).

### Evaluating replications with the 'same direction' criterion

According to our first criterion, a replication attempt is successful if the original effect and the replication effect are in the same direction. This is inclusive of original effects that are reported as a representative image without numerical values.

Among the 136 effects that were reported as being positive in the original experiments, 108 (79%) were likewise in the positive direction in the replications (*Table 1*). Moreover, the replication rate for the 101 cases where the original effect was based on a numerical result (80%) and the 35 cases where the original effect was based on a representative image (79%) were essentially the same. A weakness of the 'same direction' criterion is that it is a 'low bar' for determining replication success. If there were no true effects and all original and replication experiments were just investigating noise, the direction of original and replication effects would be random, and we would expect a 50% replication success rate. That makes 50% the lowest expected success rate with this criterion. Also, some findings have only a single direction – either the phenomenon is absent or present. As such, any detection of an effect would be labeled a success no matter the magnitude.

**Table 1.** Replication rates according to seven criteria.

| | Papers | Experiments | Effects | All outcomes |
|---|---|---|---|---|
| Total number | 23 | 50 | 158 | 188 |
| **ORIGINAL POSITIVE RESULTS** | | | | |
| *Numerical results* | | | | |
| Same direction | 17 of 19 (89%) | 26 of 35 (74%) | 80 of 101 (79%) | 95 of 116 (82%) |
| Direction and statistical significance | 8 of 19 (42%) | 17 of 33 (52%) | 42 of 97 (43%) | 44 of 112 (39%) |
| Original ES in replication CI | 5 of 19 (26%) | 3 of 33 (9%) | 17 of 97 (18%) | 26 of 112 (23%) |
| Replication ES in original CI | 5 of 19 (26%) | 11 of 33 (33%) | 42 of 97 (43%) | 50 of 112 (45%) |
| Replication ES in PI ($p_{orig}$) | 6 of 19 (32%) | 13 of 33 (39%) | 56 of 97 (58%) | 67 of 112 (60%) |
| Replication ES≥ original ES | 1 of 19 (5%) | 1 of 33 (3%) | 3 of 97 (3%) | 3 of 112 (3%) |
| Meta-analysis (p < 0.05) | 15 of 19 (79%) | 26 of 33 (79%) | 60 of 97 (62%) | 75 of 112 (67%) |
| *Representative images* | | | | |
| Same direction | 9 of 10 (90%) | 12 of 16 (75%) | 28 of 35 (80%) | 34 of 45 (76%) |
| Direction and statistical significance | 3 of 8 (40%) | 7 of 12 (58%) | 14 of 22 (64%) | 14 of 22 (64%) |
| Original image in replication CI | 5 of 7 (71%) | 3 of 11 (27%) | 10 of 21 (48%) | 10 of 21 (48%) |
| Replication effect ≥ original image | 3 of 7 (43%) | 5 of 11 (45%) | 7 of 21 (33%) | 7 of 21 (33%) |
| *Sample sizes* | | | | |
| Median [IQR] of original | 46.0 [20.0–100] | 20.0 [8.5–48.0] | 8.0 [6.0–13.0] | 8.0 [6.0–18.0] |
| Median [IQR] of replication | 50.0 [28.0–128] | 24.0 [11.5–50.0] | 12.0 [8.0–22.2] | 12.0 [8.0–18.0] |
| **ORIGINAL NULL RESULTS** | | | | |
| *Numerical results* | | | | |
| Same direction | N/A | N/A | N/A | N/A |
| Direction and statistical significance | 9 of 11 (82%) | 10 of 12 (83%) | 11 of 15 (73%) | 10 of 20 (50%) |
| Original ES in replication CI | 8 of 11 (73%) | 9 of 12 (75%) | 11 of 15 (73%) | 12 of 20 (60%) |
| Replication ES in original CI | 9 of 11 (82%) | 10 of 12 (83%) | 12 of 15 (80%) | 13 of 20 (65%) |
| Replication ES in PI ($p_{orig}$) | 9 of 11 (82%) | 10 of 12 (83%) | 12 of 15 (80%) | 14 of 20 (70%) |
| Replication ES ≤ original ES | N/A | N/A | N/A | N/A |
| Meta-analysis (p > 0.05) | 8 of 11 (73%) | 10 of 12 (83%) | 10 of 15 (67%) | 11 of 20 (55%) |
| *Representative images* | | | | |
| Same direction | N/A | N/A | N/A | N/A |
| Direction and statistical significance | 3 of 3 (100%) | 3 of 3 (100%) | 4 of 5 (80%) | 4 of 5 (80%) |
| Original image in replication CI | 1 of 3 (33%) | 1 of 3 (33%) | 3 of 5 (60%) | 3 of 5 (60%) |
| Replication effect ≤ original image | N/A | N/A | N/A | N/A |
| *Sample sizes* | | | | |
| Median [IQR] of original | 16.0 [8.0–25.0] | 12.0 [6.0–20.0] | 15.0 [7.5–31.0] | 18.0 [8.0–514] |
| Median [IQR] of replication | 24.0 [16.0–69.0] | 21.0 [8.0–54.0] | 27.0 [8.0–66.8] | 24.0 [16.0–573] |

*Table 1 continued on next page*

Table 1 continued

| | Papers | Experiments | Effects | All outcomes |
|---|---|---|---|---|

Summary of consistency between original and replication findings for original positive results (top) and null results (bottom), and by treating internal replications individually (all outcomes; column 5) and aggregated by effects (column 4), experiments (column 3), and papers (column 2). All findings coded in terms of consistency with original findings. If original results were null, then a positive result is counted as inconsistent with the original finding. For statistical significance, if original results were interpreted as a positive result but were not statistically significant at p < 0.05, then they were treated as a positive result (seven effects); likewise, if they were interpreted as a null result but were statistically significant at p < 0.05, they were treated as a null result (two effects). For original positive results, replications were deemed successful if they were statistically significant and in the same direction as the original finding; for original null results, replications were deemed successful if they were not statistically significant, regardless of direction. The 'same direction' criterion is not applicable for original null results because 'null' is an interpretation in null hypothesis significance testing and most null results still have a direction (as the effect size is almost always non-zero). Likewise, comparing direction of effect sizes is not meaningful for original null results if their variation was interpreted as noise. Mean differences were estimated from the image for original effects based on representative images. Original positive and null effects were kept separate when aggregating into experiments and papers. That is, if a single experiment had both positive and null effects, then the positive effects are summarized in 'original positive results' and the null outcomes are summarized in 'original null results'. Very similar results are obtained when alternative strategies are used to aggregate the data (see Tables S1–S3 in *Supplementary file 1*). Standardized mean difference (SMD) effect sizes are reported. CI = 95% confidence interval; PI = 95% prediction interval; ES = effect size; IQR = interquartile range.

The other 22 cases in the original experiments were reported as null effects. However, because of random error, few truly null effects are nil, literally zero, meaning that they have a direction. But, this means that there is no obvious interpretation for success or failure on the 'same direction' criterion for original null effects. Given this, we do not use the 'same direction' criterion to assess replications of original null effects.

## Evaluating replications against a null hypothesis

Null hypothesis significance testing is used to test for evidence that an observed effect size or something larger would have been unlikely to occur under the null hypothesis. For positive original results, it is straightforward to assess whether a replication effect observes a statistically significant result in the same direction as the original effect. The simplicity of this indicator has led to it being a common replication criterion despite its dichotomous nature, its dependence on the power of the replication experiment, and challenges for proper interpretation (*Andrews and Kasy, 2019*; *Camerer et al., 2016*; *Camerer et al., 2018*; *Open Science Collaboration, 2015*; *Patil et al., 2016*; *Valentine et al., 2011*). Of the 112 original effects with associated statistical significance tests, 97 were interpreted as positive effects, and 15 were interpreted as null effects (*Figure 1*).

For original positive effects, 42 of the 97 (43%) replication effects were statistically significant and in the same direction as the original effect; 48 (49%) were null results; and 7 (7%) were statistically significant in the opposite direction. Based on the power of the experiments, if the replications were all statistically consistent with the original experiments, we would expect approximately 87% of replications to be statistically significant and positive (*Mathur and VanderWeele, 2020b*), which is considerably higher than what we observed (43% [95% CI: 25%, 62%]). A sensitivity analysis that approximately accounts for possible heterogeneity within pairs also yielded a value (85%) that was considerably higher than what was observed.

For the original null effects, 11 (73%) replication effects were null results and 4 (27%) were statistically significant. Combining positive effects that remained positive and null effects that remained null, 53 of 112 (47%) of the replications were consistent with the original effects.

For cases in which the original findings were reported as representative images, we were able to conduct statistical significance tests for the replications: of the 22 effects that were positive in the original experiments, 14 (64%) replications were statistically significant in the same direction. And of the five null effects in the original experiments, 4 replications were also null (*Table 1*).

A weakness of this approach to assessing replication results is that it treats p = 0.05 as a bright-line criterion between replication success and failure. For example, if an excess of findings fell just above p = 0.05 it could indicate false negatives are present in the non-statistically significant outcomes of original positive results. p-values for non-statistically significant replication effects were widely distributed (*Figure 1—figure supplement 1*), and do not statistically differ from the approximately uniform

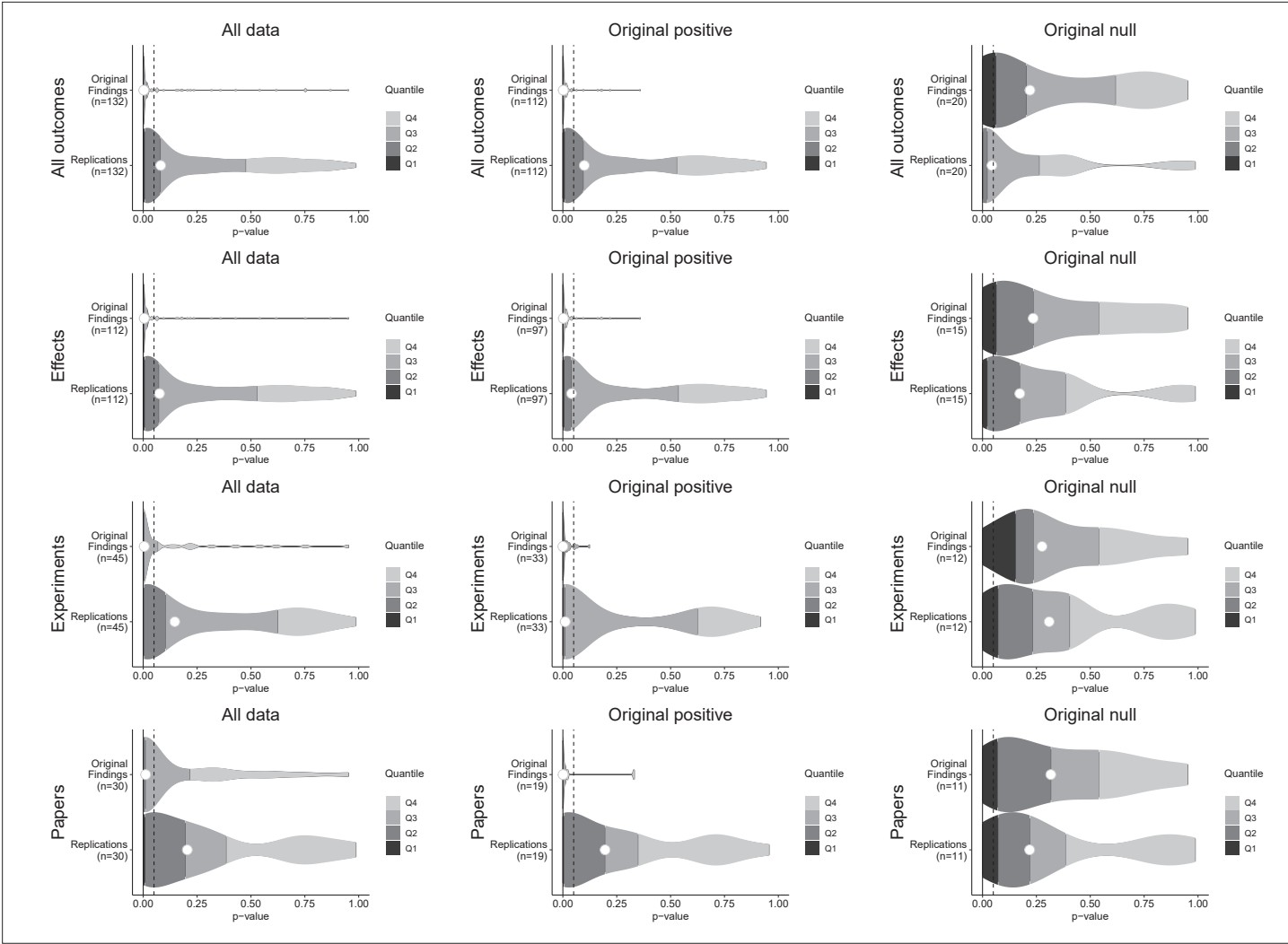

**Figure 1.** p-value density plots for original and replication results. p-alue density plots for original and replication results treating internal replications individually (top row), and aggregated by effects (second row), experiments (third row), and papers (fourth row). Left column presents all data for which p-values could be calculated for both original and replication results; the other two columns present data for when the original result was interpreted as positive (middle column) or as a null result (right column). Some original effects (n = 7) were interpreted as positive results with p-values > 0.05, and some original effects (n = 2) were interpreted as null results with p-values < 0.05. Replication p-values ignore whether the result was in the same or opposite direction as the original result (n = 7 effects had p-values < 0.05 in the opposite direction as the original effect).

The online version of this article includes the following figure supplement(s) for figure 1:

**Figure supplement 1.** p-value distributions for original and replication effects.

distribution that would be expected if all were true null results whether examining the findings that had p-values for both original and replication effects (Fisher's exact test: $\chi^2(118) = 135.7$, p = 0.127), or also including the replication effects for which the original effects were based on a representative image ($\chi^2(138) = 155.1$, p = 0.152). Therefore, we cannot reject the hypothesis that the observed null effects come from a population of true negatives.

## Comparing original effect size with the 95% confidence interval of the replication effect size

Another approach based on the logic of null hypothesis significance testing is to assess whether the original effect size is contained within the 95% confidence interval of the replication effect size. In this approach the null hypothesis is the original effect size, and we are testing if the replication effect size is significantly different. We found that 17 of the 97 (18%) original positive effect sizes were in the 95%

confidence interval of the replication effect size, as were 11 of the 15 (73%) original null effect sizes. Therefore, according to this criterion, 75% of original effect sizes were inconsistent with the replications, even if both observed an effect in the same direction. Note that the precision of the replication estimate is influenced by the sample size, which was larger in the replications than in the original experiments for both positive effects (12.0 vs. 8.9 for the median) and null effects (27.0 vs. 15.0).

This criterion can also be applied to cases in which the original experiment reported only a representative image with an effect size that could be estimated from that image. Of these, 10 of the 21 (48%) original positive effect sizes, and 3 of the 5 (60%) original null effect sizes, were in the 95% confidence interval of the replication effect size, meaning that half of the original effect sizes were in the confidence interval of the replication effect sizes, and half of the original images were inconsistent with the replications. Combining all numerical and image-only data, 41 of 138 (30%) replications were consistent with original effects on this criterion.

## Comparing the replication effect size with the 95% confidence interval of the original effect size

A complementary criterion is to assess whether the replication effect size falls within the 95% confidence interval of the original effect size. When the original effect was positive, 42 of the 97 (43%) replication effect sizes were within the 95% confidence interval of original effect size; and when the original effect was null, 12 of the 15 (80%) replication effect sizes were within the 95% confidence interval of the original effect sizes. This success rate is low but it is almost double the rate reported for a seemingly similar approach in the previous section. This is attributable to the smaller sample sizes in the original experiments leading to wider confidence intervals, thus making it 'easier' for the replication to achieve an effect size that counts as a success.

A more complete picture of the consistency between the original findings and the replications in the null hypothesis significance testing framework can be obtained by combining the three criteria we have just discussed. Were the original results and the replications consistent on zero, one, two, or all three of these criteria? For the 97 effects that were positive in the original experiments, we find that just 13 were successful on all three criteria, 18 were successful on two, 26 were successful on one, and 40 failed to replicate on all three criteria. For the 15 effects that were null in the original experiments, eight were successful on all three criteria, four were successful on two, two were successful on one, and only one failed on all three criteria (see *Table 2* and Tables S4–S6 in *Supplementary file 1*).

## Comparing the replication effect size with the 95% prediction interval of the original effect size

A 95% prediction interval is the range of values inside which a future observation will fall with 95% probability, given what has been observed already. Prediction intervals are sometimes preferred over confidence intervals when presenting new results because they do not assume that the future observation has infinite sample size. As a consequence, they more appropriately represent the (greater) uncertainty around the future estimate.

However, as a criterion for evaluating replication success or failure, prediction intervals are more liberal than criteria based on confidence intervals. If, for example, the original finding was close to p = 0.05, then the prediction interval will often overlap with zero. If the true effect size is near zero, a replication might never provide evidence inconsistent with the prediction interval unless random error leads to the effect size being estimated in the opposite direction of the original finding. In other words, somewhat ironically, the more uncertain an original finding, the harder it is for a replication to provide disconfirming evidence. Nevertheless, the prediction interval has been used in at least one case to estimate replication success (*Patil et al., 2016*).

For the 97 effects that were positive in the original experiments, 56 effects (58%, 95% CI: [44%, 72%]) could be considered successful according to this criterion (*Table 1*). A sensitivity analysis that approximately accounts for possible heterogeneity within pairs yields a higher value (65%, 95% CI: [51%, 79%]). And for the 15 effects that were null in the original experiments, 12 effects (80%) could be considered successful. Combining these results, 68 of 112 (61%) replications were successful according to prediction interval criterion.

Related to prediction intervals, the degree of statistical inconsistency between each replication and the corresponding original effect can be represented with a metric called $p_{orig}$, which is a p-value for

**Table 2.** Replication rates according to three criteria involving null hypothesis significance testing.

| | Papers | | Experiments | | Effects | | All outcomes | |
|---|---|---|---|---|---|---|---|---|
| Total number | 23 | | 50 | | 158 | | 188 | |
| **ORIGINAL POSITIVE RESULTS** | | | | | | | | |
| Succeeded on all three criteria | 2 | 11% | 2 | 6% | 13 | 13% | 20 | 18% |
| [1]Failed only on significance and direction | 2 | 11% | 1 | 3% | 4 | 4% | 6 | 5% |
| [2]Failed only on original in replication confidence interval | 1 | 5% | 5 | 15% | 14 | 14% | 10 | 9% |
| [3]Failed only on replication in original confidence interval | 0 | 0% | 0 | 0% | 0 | 0% | 0 | 0% |
| Failed only on [1] and [2] | 0 | 0% | 3 | 9% | 11 | 11% | 14 | 13% |
| Failed only on [2] and [3] | 5 | 26% | 10 | 30% | 15 | 15% | 14 | 13% |
| Failed only on [1] and [3] | 1 | 5% | 0 | 0% | 0 | 0% | 0 | 0% |
| Failed on all three criteria [1], [2], and [3] | 8 | 42% | 12 | 36% | 40 | 41% | 48 | 43% |
| Total | 19 | | 33 | | 97 | | 112 | |
| **ORIGINAL NULL RESULTS** | | | | | | | | |
| Succeeded on all three criteria | 6 | 55% | 7 | 58% | 8 | 53% | 7 | 35% |
| [1]Failed only on significance and direction | 2 | 18% | 2 | 17% | 3 | 20% | 5 | 25% |
| [2]Failed only on original in replication confidence interval | 1 | 9% | 1 | 8% | 1 | 7% | 1 | 5% |
| [3]Failed only on replication in original confidence interval | 0 | 0% | 0 | 0% | 0 | 0% | 0 | 0% |
| Failed only on [1] and [2] | 0 | 0% | 0 | 0% | 0 | 0% | 0 | 0% |
| Failed only on [2] and [3] | 2 | 18% | 2 | 17% | 2 | 13% | 2 | 10% |
| Failed only on [1] and [3] | 0 | 0% | 0 | 0% | 0 | 0% | 0 | 0% |
| Failed on all three criteria [1], [2], and [3] | 0 | 0% | 0 | 0% | 1 | 7% | 5 | 25% |
| Total | 11 | | 12 | | 15 | | 20 | |

Number of replications that succeeded or failed to replicate results in original experiments according to three criteria within the null hypothesis significance testing framework: statistical significance ($p < 0.05$) and same direction; original effect size inside 95% confidence interval of replication effect size using standardized mean difference (SMD) effect sizes; replication effect size inside 95% confidence interval of original effect size using SMD effect sizes. Data for original positive results and original null results are shown separately, as are data for all outcomes and aggregated by effect, experiment, and paper. Very similar results are obtained when alternative strategies are used to aggregate the data (see Tables S4–S6 in **Supplementary file 1**).

the hypothesis that the original and the replication had the same population effect size (*Mathur and VanderWeele, 2020b*). $p_{orig}$ thus assesses whether each replication effect was similar to the corresponding original effect, with small values of $p_{orig}$ indicating less similarity and larger values indicating more similarity. For original positive effects, the median $p_{orig}$ was 0.064, suggesting some evidence for inconsistency on average. Of the 97 original positive effects, 42% (95% CI: [28%, 56%]) had $p_{orig} < 0.05$, and 14% (95% CI: [5%, 24%]) had $p_{orig} < 0.005$ (*Benjamin et al., 2018*).

We then aggregated the values of $p_{orig}$ using the harmonic mean p-value to test the global null hypothesis that none of the pairs were statistically inconsistent (*Wilson, 2019*), yielding an aggregated $p_{orig}$ of 0.0005, which is strong evidence of some inconsistency in the tested pairs. The aggregated value of $p_{orig}$ accommodates correlations among p-values due to nested data. In a sensitivity analysis that additionally accounted for possible effect heterogeneity within each of the 97 original-replication pairs, the median $p_{orig}$ was 0.087: 35% of effects (95% CI: [21%, 49%]) had $p_{orig} < 0.05$, and 12% of effects (95% CI: [2%, 23%]) had $p_{orig} < 0.005$.

**Table 3.** Comparing effect sizes in the original results and the replications.

| | Papers | Experiments | Effects | All outcomes |
|---|---|---|---|---|
| **ORIGINAL POSITIVE RESULTS** | | | | |
| Number of outcomes | 19 | 33 | 97 | 112 |
| Mean (SD) original experiment effect size | 7.35 (18.77) | 6.36 (14.62) | 6.15 (12.39) | 5.56 (11.63) |
| Median [IQR] original experiment effect size | 2.07 [1.68–5.03] | 2.45 [1.42–4.58] | 2.96 [1.71–5.70] | 2.57 [1.60–5.49] |
| Mean (SD) replication experiment effect size | 1.38 (2.02) | 1.55 (3.31) | 1.37 (3.01) | 1.30 (2.83) |
| Median [IQR] replication experiment effect size | 0.53 [0.18–1.80] | 0.37 [0.10–1.31] | 0.43 [0.15–2.06] | 0.47 [0.17–1.67] |
| Meta-analytic mean (SD) estimate | 1.68 (1.81) | 1.79 (2.90) | 1.66 (2.47) | 1.61 (2.32) |
| Meta-analytic median [IQR] estimate | 0.98 [0.57–2.20] | 1.00 [0.28–2.03] | 0.92 [0.36–2.43] | 1.05 [0.36–2.11] |
| *Sample sizes* | | | | |
| Median [IQR] of original | 46.0 [20.0–100] | 24.0 [9.0–48.0] | 8.0 [6.0–13.0] | 8.5 [6.0–18.0] |
| Median [IQR] of replication | 50.0 [28.0–128] | 32.0 [12.0–50.0] | 12.0 [8.0–23.0] | 12.0 [8.0–18.0] |
| **ORIGINAL NULL RESULTS** | | | | |
| Number of outcomes | 11 | 12 | 15 | 20 |
| Mean (SD) original experiment effect size | 0.70 (0.64) | 0.72 (0.61) | 0.63 (0.59) | 0.51 (0.55) |
| Median [IQR] original experiment effect size | 0.61 [0.15–1.03] | 0.68 [0.15–1.03] | 0.61 [0.16–0.97] | 0.18 [0.15–0.79] |
| Mean (SD) replication experiment effect size | –0.08 (0.75) | –0.02 (0.74) | 0.02 (0.69) | 0.01 (0.86) |
| Median [IQR] replication experiment effect size | 0.13 [-0.27–0.24] | 0.13 [-0.23–0.39] | 0.16 [-0.24–0.47] | 0.16 [-0.21–0.39] |
| Meta-analytic mean (SD) estimate | 0.20 (0.31) | 0.25 (0.34) | 0.24 (0.34) | 0.20 (0.39) |
| Meta-analytic median [IQR] estimate | 0.17 [0.06–0.40] | 0.23 [0.07–0.43] | 0.16 [0.06–0.44] | 0.16 [0.07–0.43] |
| *Sample sizes* | | | | |
| Median [IQR] of original | 16.0 [8.0–25.0] | 12.0 [7.0–22.5] | 18.0 [8.0–32.0] | 19.0 [11.0–514] |
| Median [IQR] of replication | 24.0 [16.0–69.0] | 22.5 [8.0–61.5] | 30.0 [12.0–72.5] | 27.0 [17.5–573] |

Comparing original effect sizes and effect sizes in the replications for original positive results (top) and null results (bottom) when treating internal replications individually (all outcomes; column 5) and aggregated by effects (column 4), experiments (column 3), and papers (column 2). The mean and median of the effect sizes in the original results were considerably larger than those for the replications. SD = standard deviation; IQR = interquartile range.

## Comparing effect sizes in the original experiments and the replications

Another way to assess replications is to compare the original effect size and the replication effect size. For the 97 effects that were positive in the original experiments, the effect size was lower in 94 (97%) of the replications (*Table 1*): if the original effect sizes were accurately estimated, one would expect this percentage to be about 50%, and the probability of 94 of the 97 replication effect sizes being lower than the original effect sizes would be vanishingly low (binomial test: $p = 1.92 \times 10^{-24}$). And for the 21 cases in which the original evidence for a positive effect was a representative image, the effect size in the replication was smaller than the effect size estimated for the original in 14 cases (67%). Combining these results, the effect sizes in the replications were smaller than the effect sizes in the original findings in 108 of 118 (92%) cases.

We also compared the mean and median values of the effect sizes for positive effects (*Table 3*): in both cases the value was considerably larger for the original effect. Comparing means, the value for the original effects was 6.15 (SD = 12.39, 95% CI: [1.83, 10.47]), and the value for the replications was

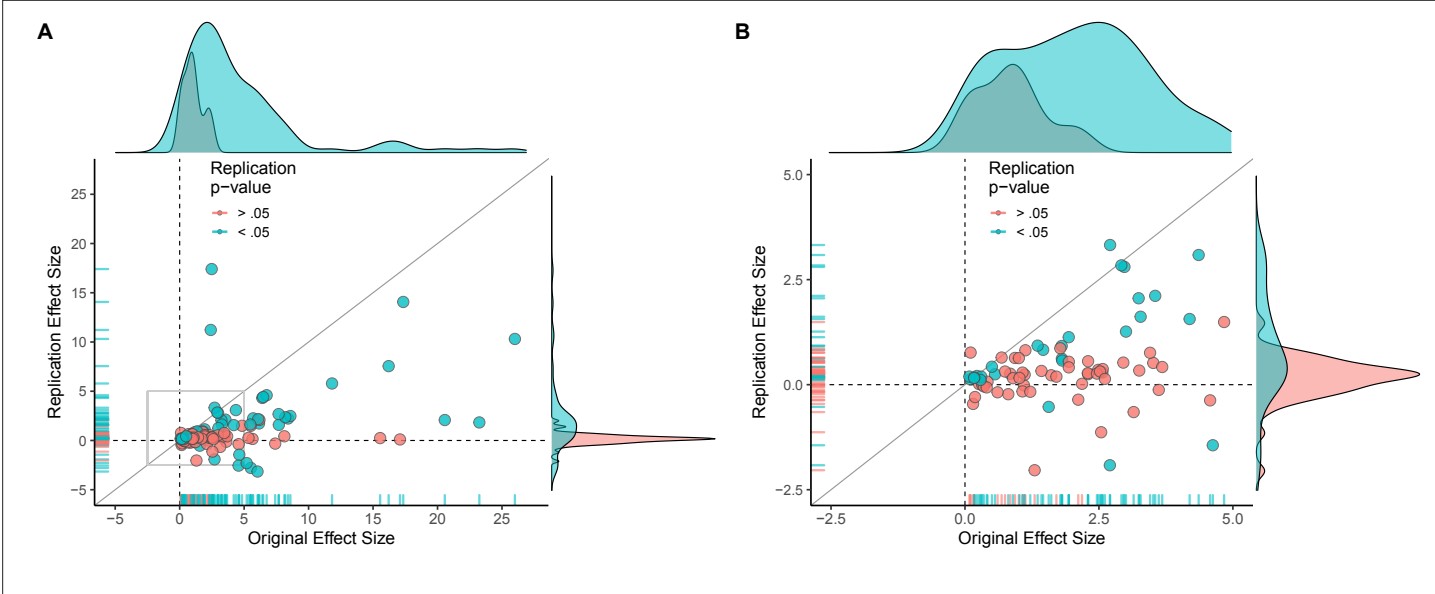

**Figure 2.** Replication effect sizes compared with original effect sizes. (**A**) Graph in which each circle represents an effect for which an SMD effect size could be computed for both the original effect and the replication (n = 110). Blue circles indicate effects for which p < 0.05 in the replication, and red circles indicate p > 0.05. Two effects for which the original effects size was >80 are not shown. The median effect size in the replications was 85% smaller than the median effect size in the original experiments, and 97% of replication effect sizes were smaller than original effect sizes (below the gray diagonal line). (**B**) An expanded view of panel A for effect sizes < 5 (gray outline in panel A). SMD: standardized mean difference.

The online version of this article includes the following figure supplement(s) for figure 2:

**Figure supplement 1.** Replication effect sizes compared with original effect sizes for all effects (treating internal replications individually).

**Figure supplement 2.** Replication effect sizes compared with original effect sizes for experiments (combining effects).

**Figure supplement 3.** Replication effect sizes compared with original effect sizes for papers (combining experiments).

1.37 (SD = 3.01, 95% CI: [0.42, 2.32]). Comparing medians, the value for the original effects was 2.96 (interquartile range [IQR] = 1.71–5.70), and the value for the replications was 0.43 (IQR = 0.15–2.06).

The pattern was similar when we compared mean and median values for null effects. Comparing means, the value for the original effects was 0.63 (SD = 0.59), and the value for replications was 0.02 (SD = 0.69). Comparing medians, the value for the original effects was 0.61 (IQR = 0.16–0.97), and the value for replications was 0.16 (IQR = –0.24–0.47).

Although the original and replication effect sizes had very different effect magnitudes, larger effect sizes in the original results tended to be associated with larger effect sizes in the replication (Spearman's r = 0.47, p = $1.83 \times 10^{-7}$; *Figure 2*). This indicates that observed effect sizes are not all random, and that some findings retain their rank ordering in effect size, despite the clear differences between the original and replication effect sizes. To illustrate the comparability of these findings across different levels of aggregation, *Figure 3* presents density plots of original effect sizes compared to replication effect sizes by individual outcomes, effects, experiments, and papers.

## Combining the original and replication effect sizes

Combining the original and replication findings provides an assessment of the cumulative evidence for a phenomenon. In general, cumulative or meta-analytic evidence obtained from multiple independently conducted experiments provides a better basis for assessing the reliability of findings than evidence from a single experiment. However, the credibility of such results is contingent on a lack of selective reporting or on the ability to effectively correct for missing evidence (*McShane et al., 2016*; *Stanley and Doucouliagos, 2014*). If, for example, original experiments were influenced by publication bias, with null results being ignored at greater rates, then the meta-analytic evidence would be biased. The use of preregistration and complete outcome reporting in the project eliminated the possibility of publication bias in the replication experiments (*Errington et al., 2021b*), but it may be present in the original experiments. Nevertheless, we combined the two sets of results by weighting

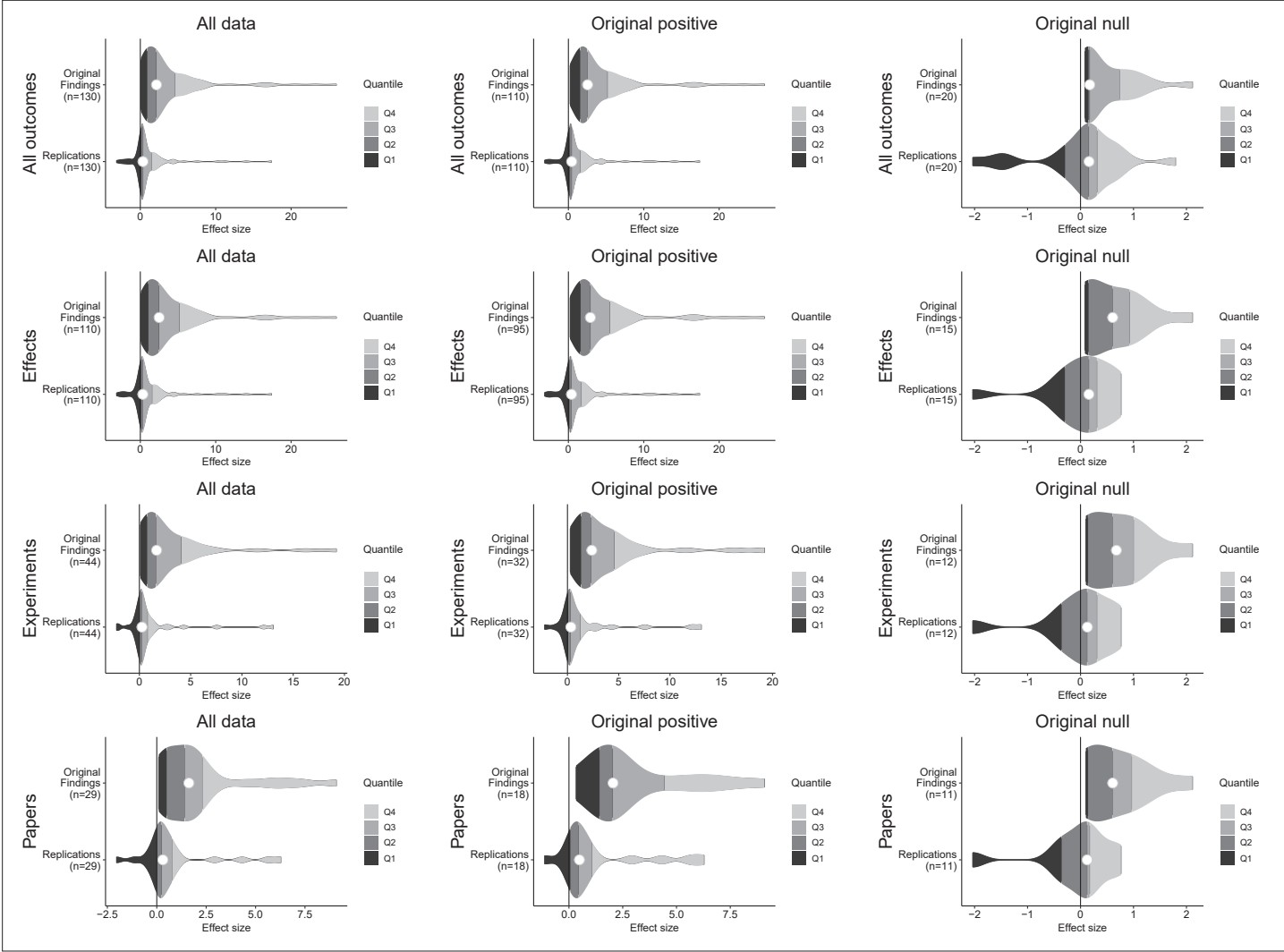

**Figure 3.** Effect size density plots for original and replication results. Effect size density plots for original and replication findings for all results treating internal replications individually (top row) and aggregated by effects (second row), experiments (third row), and papers (fourth row). Left column presents all data for which SMD effect sizes could be calculated for both original and replication results; the other two columns present data for when the original result was interpreted as positive (middle column) or as a null result (right column). Effect sizes > 80 (two for all outcomes and effects, and one for experiments and papers) are not shown.

The online version of this article includes the following figure supplement(s) for figure 3:

**Figure supplement 1.** Effect size distributions for original and replication effects.

each finding by the inverse of its variance to estimate the effect size and effect precision. Using a fixed-effect model for each original-replication pair where the original result was positive, 60 of the 97 effects (62%) were statistically significant at $p < 0.05$ (*Table 1*), and 39 effects (40% of the total) were statistically significant according to the stricter criterion of $p < 0.005$. *Table 3* reports mean and median values for the original effect size, the replication effect size, and the meta-analytic combination of the two. According to the combined results the mean effect size was 1.66 (95% CI: [0.92, 2.41]).

For cases in which the original was a null effect, 10 of the 15 (67%) meta-analytic effects were likewise null ($p > 0.05$), meaning that combining the data led to a third of cases showing statistically significant effects, even though the original reported a null finding. This can occur when the original experiment was underpowered to detect the true effect size, but combining data increases precision to detect smaller effects. This can be important when, for example, an experiment is evaluating whether an intervention increases toxicity. An original null result based on a small sample may provide

misplaced confidence in safety. Considering all data, 70 of 112 (63%) of meta-analytic combinations showed effects consistent with the effect reported for the original experiment.

## Comparing animal vs. non-animal experiments

Animal experiments have special significance in understanding biological mechanisms and in translating basic science into potential clinical application. We explored whether the patterns of replicability differed between the animal and non-animal experiments included in this meta-analysis (*Table 4*). Descriptively, animal experiments with positive effects were less likely to replicate than non-animal experiments with positive effects on every replication criterion. For example, 12% of replication effects were in the same direction as the original and statistically significant for animal experiments, compared with 54% for non-animal experiments. Likewise, 44% of replication effects were in the 95% prediction interval for animal experiments, compared with 63% for non-animal experiments.

We used multi-level models to explore the association of five possible moderators with the replication rate: (1) animal experiments vs. non-animal (i.e., in vitro) experiments; (2) the use of contract research organizations to conduct replications; (3) the use of academic research core facilities to conduct replications; (4) whether the original authors shared materials with the replicating labs; (5) the quality of methodological clarifications made by the original authors upon request from the replicating labs (*Errington et al., 2021b*). None of the five moderators showed a consistent, significant association with replication success (see Table S7 in *Supplementary file 1*), though the moderators were variably correlated with one another (ranging from r = –0.68 to r = 0.53; *Figure 4*). We cannot say whether any of these moderators influence replication success in general, but this analysis suggests that they do not account for much variation in replication success in this sample.

Other factors have been identified that could improve replicability, such as blinding, randomization, and sample size planning (*Landis et al., 2012*; *Macleod and Mohan, 2019*). However, these aspects were very rarely reported in the original experiments so they could not be examined as candidate moderators. For example, for the 36 animal effects across 15 experiments, none of the original experiments reported blinding, one experiment (for two effects) reported randomization, and none reported determining sample size a priori. By comparison, the replications reported blinding for five experiments (11 effects), randomization for 13 experiments (28 effects), and all 15 experiments (36 effects) reported calculating sample size a priori.

The multi-level analysis does not support the conclusion that there is a meaningful difference in the replication rates of animal and non-animal experiments. As can be seen in *Table 5*, median effect sizes were 84% smaller than the original findings for animal replications, and 78% smaller for non-animal replications. The reason that animal experiments had such a low replication rate, particularly according to the statistical significance criterion (12%), is that the effect sizes in the original experiments (Mdn = 1.61) were notably smaller than the effect sizes in the original non-animal experiments (Mdn = 3.65; *Figure 5*). In sum, original findings with smaller effect sizes were less likely to replicate than original findings with larger effect sizes, and animal experiments tended to have smaller effect sizes than non-animal experiments. In other words, when seeking to predict if a replication will be successful it is more useful to know the original effect size than to know whether the original experiment was an animal experiment or not.

## Summarizing replications across five criteria

The criteria described above returned a range of replication rates due to the different assumptions made by each, particularly how they handle the estimation of uncertainty. To provide an overall picture, we combined the replication rates by five of these criteria, selecting criteria that could be meaningfully applied to both positive and null effects, which meant excluding the 'same direction' and 'comparing effect size' criteria, as neither works for null effects.

For replications of original positive effects, 13 of 97 (13%) replications succeeded on all five criteria, 15 succeeded on four, 11 succeeded on three, 22 failed on three, 15 failed on four, and 21 (22%) failed on all five (*Table 6* and *Figure 6*). For original null effects, 7 of 15 (47%) replications succeeded on all five criteria, 2 succeeded on four, 3 succeeded on three, 0 failed on three, 2 failed on four, and 1 (7%) failed on all five. If we consider a replication to be successful overall if it succeeded on a majority of criteria (i.e., three or more), original null effects (80%) were twice as likely to replicate as original positive effects (40%). Combining positive and null effects, 51 of 112 (46%) replications succeeded on

**Table 4.** Replication rates for animal and non-animal experiments.

| | Animal | Non-animal | Total |
|---|---|---|---|
| Total number of effects | 36 | 122 | 158 |
| **ORIGINAL POSITIVE EFFECTS** | | | |
| *Numerical results* | | | |
| Same direction | 17 of 27 (63%) | 63 of 74 (85%) | 80 of 101 (79%) |
| Direction and statistical significance | 3 of 25 (12%) | 39 of 72 (54%) | 42 of 97 (43%) |
| Original ES in replication CI | 4 of 25 (16%) | 13 of 72 (18%) | 17 of 97 (18%) |
| Replication ES in original CI | 9 of 25 (36%) | 33 of 72 (46%) | 42 of 97 (43%) |
| Replication ES in PI ($p_{orig}$) | 11 of 25 (44%) | 45 of 72 (63%) | 56 of 97 (58%) |
| Replication ES≥ original ES | 0 of 25 (0%) | 3 of 72 (4%) | 3 of 97 (3%) |
| Meta-analysis (p < 0.05) | 13 of 25 (52%) | 47 of 72 (65%) | 60 of 97 (62%) |
| *Representative images* | | | |
| Same direction | 1 of 4 (25%) | 27 of 31 (87%) | 28 of 35 (80%) |
| Direction and statistical significance | 0 of 2 (0%) | 14 of 20 (70%) | 14 of 22 (64%) |
| Original image in replication CI | 0 of 1 (0%) | 10 of 20 (50%) | 10 of 21 (48%) |
| Replication effect ≥ original image | 0 of 1 (0%) | 7 of 20 (35%) | 7 of 21 (33%) |
| *Sample sizes* | | | |
| Median [IQR] of original | 14.0 [10.0–20.0] | 7.0 [6.0–11.2] | 8.0 [6.0–13.0] |
| Median [IQR] of replication | 15.0 [13.0–21.8] | 10.0 [8.0–22.0] | 12.0 [8.0–22.2] |
| **ORIGINAL NULL EFFECTS** | | | |
| *Numerical results* | | | |
| Same direction | N/A | N/A | N/A |
| Direction and statistical significance | 4 of 5 (80%) | 7 of 10 (70%) | 11 of 15 (73%) |
| Original ES in replication CI | 4 of 5 (80%) | 7 of 10 (70%) | 11 of 15 (73%) |
| Replication ES in original CI | 5 of 5 (100%) | 7 of 10 (70%) | 12 of 15 (80%) |
| Replication ES in PI ($p_{orig}$) | 5 of 5 (100%) | 7 of 10 (70%) | 12 of 15 (80%) |
| Replication ES≤ original ES | N/A | N/A | N/A |
| Meta-analysis (p > 0.05) | 3 of 5 (60%) | 7 of 10 (70%) | 10 of 15 (67%) |
| *Representative images* | | | |
| Same direction | N/A | N/A | N/A |
| Direction and statistical significance | N/A | 7 of 5 (80%) | 4 of 5 (80%) |
| Original image in replication CI | N/A | 3 of 5 (60%) | 3 of 5 (60%) |
| Replication effect ≤ original image | N/A | N/A | N/A |
| *Sample sizes* | | | |
| Median [IQR] of original | 21.0 [20.0–30.0] | 8.0 [5.0–266] | 15.0 [7.5–31.0] |
| Median [IQR] of replication | 35.0 [30.0–61.0] | 16.0 [7.0–604] | 27.0 [8.0–66.8] |

*Table 4 continued on next page*

*Table 4 continued*

| | Animal | Non-animal | Total |
| --- | --- | --- | --- |

Comparing replication rates for animal experiments (column 2) and non-animal experiments (column 3) according to the seven criteria used in *Table 1*. For statistical significance, if original effects were interpreted as a positive effect but were not significant at p < 0.05, then they were treated as a positive effect (7 cases), and likewise if they were interpreted as a null effect but were significant at p < 0.05 they were treated as a null effect (3 cases). Standardized mean difference (SMD) effect sizes are reported. CI = 95% confidence interval; PI = 95% prediction interval; ES = effect size; IQR = interquartile range.

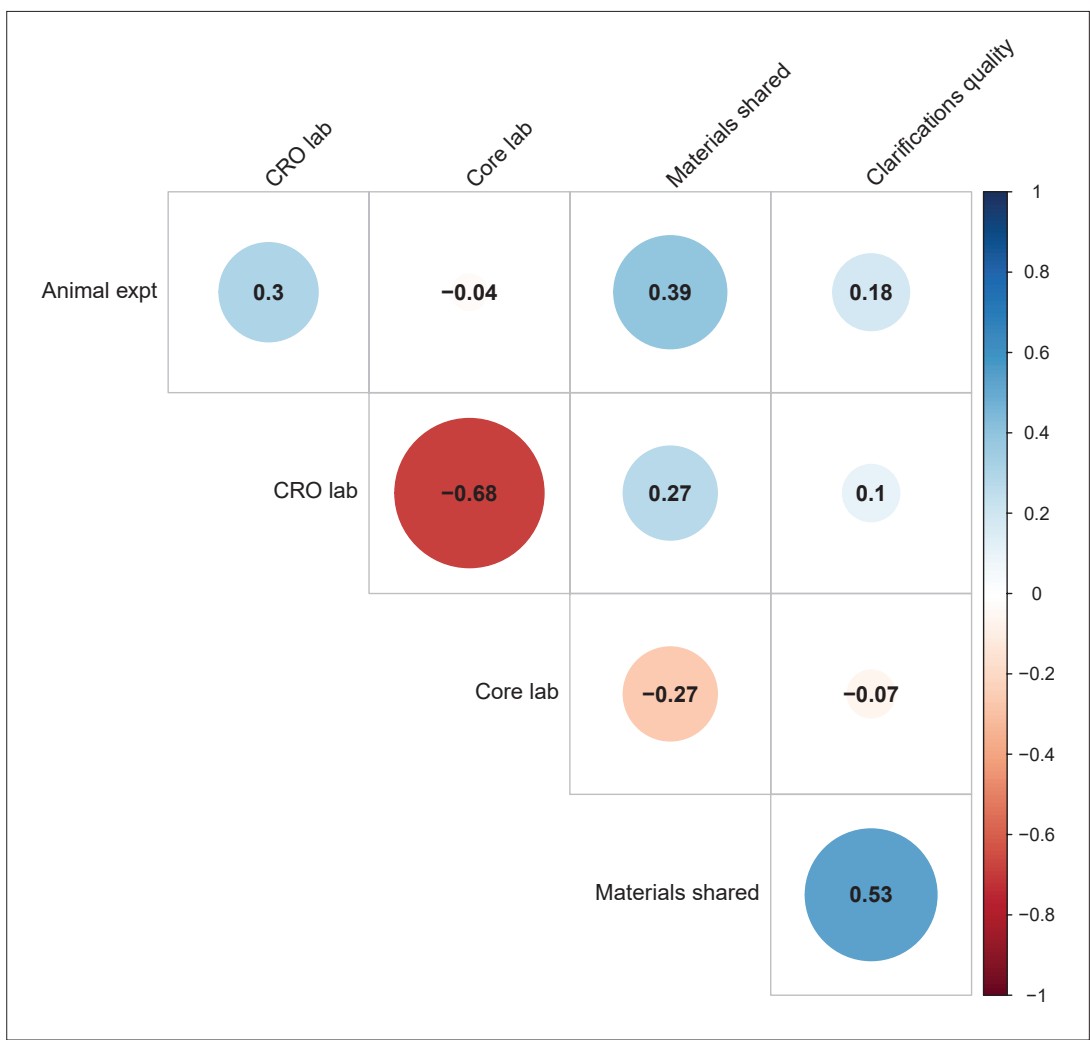

**Figure 4.** Correlations between five candidate moderators. Point-biserial correlations among five candidate moderators for predicting replication success for the 97 original positive effects with replication pairs. The five moderators were: (i) animal experiments vs. non-animal (i.e., in vitro) experiments (animal expt); (ii) the use of contract research organizations to conduct replications (CRO lab); (iii) the use of academic research core facilities to conduct replications (core lab); (iv) whether the original authors shared materials with the replicating labs (materials shared); (v) the quality of methodological clarifications made by the original authors (clarifications quality); see Materials and methods for more details. Correlations are color-coded (blue = positive; red = negative; see color bar), with the size of the circle being proportional to the magnitude of the correlation. None of the five moderators showed a consistent, significant association with replication rate (see Table S7 in ***Supplementary file 1***).

**Table 5.** Effect sizes for animal and non-animal experiments.

| | Animal | Non-animal | Total |
|---|---|---|---|
| **ORIGINAL POSITIVE EFFECTS** | | | |
| Number of outcomes | 25 | 72 | 97 |
| Mean (SD) original experiment effect size | 1.88 (1.61) | 7.63 (14.07) | 6.15 (12.39) |
| Median [IQR] original experiment effect size | 1.61 [0.81–2.30] | 3.65 [2.45–6.43] | 2.96 [1.71–5.70] |
| Mean (SD) replication experiment effect size | 0.19 (0.50) | 1.78 (3.39) | 1.37 (3.01) |
| Median [IQR] replication experiment effect size | 0.25 [−0.06–0.41] | 0.79 [0.20–2.27] | 0.43 [0.15–2.06] |
| Meta-analytic mean (SD) estimate | 0.65 (0.54) | 2.02 (2.77) | 1.66 (2.47) |
| Meta-analytic median [IQR] estimate | 0.83 [0.11–1.05] | 1.06 [0.46–2.79] | 0.92 [0.36–2.43] |
| **ORIGINAL NULL EFFECTS** | | | |
| Number of outcomes | 5 | 10 | 15 |
| Mean (SD) original experiment effect size | 0.34 (0.29) | 0.78 (0.65) | 0.63 (0.59) |
| Median [IQR] original experiment effect size | 0.19 [0.10–0.61] | 0.84 [0.17–1.08] | 0.61 [0.16–0.97] |
| Mean (SD) replication experiment effect size | 0.21 (0.48) | −0.08 (0.78) | 0.02 (0.69) |
| Median [IQR] replication experiment effect size | 0.13 [−0.18–0.65] | 0.16 [−0.27–0.28] | 0.16 [−0.24–0.47] |
| Meta-analytic mean (SD) estimate | 0.21 (0.31) | 0.25 (0.37) | 0.24 (0.34) |
| Meta-analytic median [IQR] estimate | 0.12 [0.04–0.37] | 0.17 [0.10–0.45] | 0.16 [0.06–0.44] |

Comparing original and replication effect sizes (means and medians) for animal experiments (column 2) and non-animal experiments (column 3), along with meta-analytic means and medians for the effect size obtained by combining data from the original effects and the replications. SD = standard deviation; IQR = interquartile range.

more criteria than they failed, and 61 (54%) replications failed on more criteria than they succeeded. We also found these five criteria to be positively correlated with one another ranging from 0.15 to 0.78 and a median of 0.345 suggesting that they provide related but distinct information (*Figure 7*).

The 'same direction' and 'comparing effect size' criteria were not included as neither works for null effects. Also, these two criteria cannot be directly compared with the other five criteria because they both have a minimum replication success rate of 50% under ordinary assumptions, compared with 0% for the other criteria. For example, if all the original effects were due to noise, then the 'same direction' criterion would return a value of 50% for the replication rate indicating the worst possible performance. The observation that 79% of replications were in the same direction as the original effect indicates some signal being detected. Conversely, if all the original and replication effect sizes were equivalent and estimated without bias, then the 'comparing effect size' criterion would return a value of 50% indicating the best possible performance. The observation that just 3% of replications had larger effect sizes than original positive effects indicates that the original effect sizes were overestimated.

## Discussion

We used seven criteria to assess the replicability of 158 effects in a selection of 23 papers reporting the results of preclinical research in cancer biology. Across multiple criteria, the replications provided weaker evidence for the findings than the original papers. For original positive effects that were reported as numerical values, the median effect size for the replications was 0.43, which was 85% smaller than the median of the original effect sizes (2.96). And although 79% of the replication effects were in the same direction as the original finding (random would be 50%), 92% of replication effect sizes were smaller than the original (combining numeric and representative images). Across five

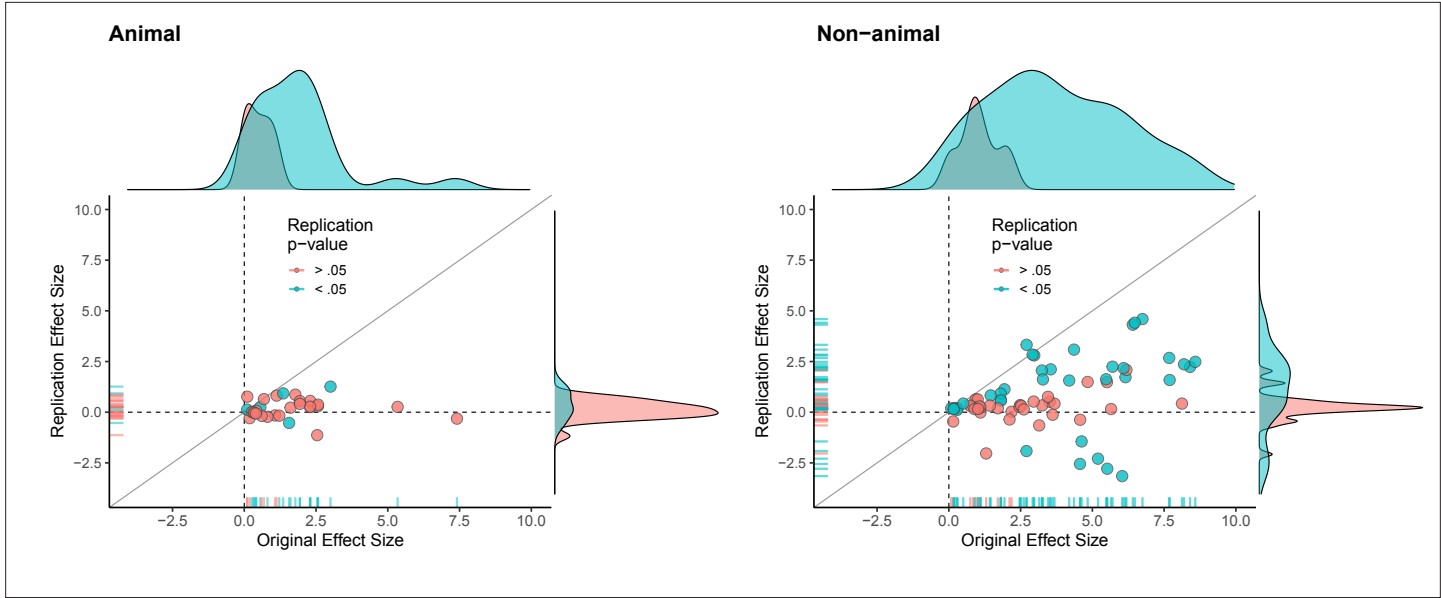

**Figure 5.** Replication effect sizes compared with original effect sizes for animal and non-animal experiments. Graphs for animal experiments (n = 30 effects; left) and non-animal experiments (n = 70 effects; right) in which each circle represents an effect for which an SMD effect size could be computed for both the original effects and the replication. Blue circles indicate effects for which p < 0.05 in the replication, and red circles indicate p > 0.05. Animal experiments were less likely to replicate than non-animal experiments and this may be a consequence of animal experiments eliciting smaller effect sizes on average than non-animal experiments (see main text for further discussion). Twelve effects in the non-animal experiments for which the original effects size was >10 are not shown. SMD: standardized mean difference.

dichotomous criteria for assessing replicability, original null results were twice as likely as original positive results to mostly replicate successfully (80% vs. 40%). Combining original positive and null effects for each of the five criteria, the replication success rates were 47% for same direction and statistical significance, 25% for the original effect size being inside the 95% confidence interval (CI) of the replication, 48% for the replication effect size being inside the 95% CI of the original, 61% for the replication effect size being inside the 95% prediction interval, and 63% for a criterion based on a meta-analytic combination of the data from the original experiment and the replication. Replication rates were relatively consistent whether examining the effects in isolation, combining across internal replications, combining across all the effects of each experiment, or combining across all the experiments of each paper. Animal and non-animal replications both had similarly weaker effect sizes compared to original findings, but animal experiments were much less likely to replicate (probably because the original effect size tended to be smaller in animal experiments). In this section we discuss some of the implications of our results.

## What does a failure to replicate mean?

A single failure to replicate a finding does not render a verdict on its replicability or credibility. A failure to replicate could occur because the original finding was a false positive. Indeed, there is accumulating evidence of the deleterious impacts of low power and small sample sizes, ignoring null results, failures of transparency of methodology, failing to publish all experimental data collected, and questionable research practices such as p-hacking on the inflation of false positives in the published literature (*Casadevall and Fang, 2012*; *Chalmers et al., 2014*; *Gelman and Loken, 2013*; *Greenwald, 1975*; *Ioannidis, 2005*; *John et al., 2012*; *Kaplan and Irvin, 2015*; *Landis et al., 2012*; *Macleod et al., 2014*; *Macleod et al., 2015*; *van der Naald et al., 2020*; *Rosenthal, 1979*; *Simmons et al., 2011*). This evidence suggests that published findings might often be false positives or have exaggerated effect sizes, potentially adding noise and friction to the accumulation of knowledge.

A failure to replicate could also occur because the replication was a false negative. This can occur if the replication was underpowered or the design or execution was flawed. Such failures are uninteresting but important. Minimizing them requires attention to quality and rigor. We contracted independent laboratories with appropriate instrumentation and expertise to conduct the experiments. This

**Table 6.** Assessing replications of positive and null results across five criteria.

| | Papers | | Experiments | | Effects | | All outcomes | |
|---|---|---|---|---|---|---|---|---|
| **ORIGINAL POSITIVE RESULTS** | | | | | | | | |
| Successful replication on all five criteria | 2 | 11% | 2 | 6% | 13 | 13% | 20 | 18% |
| Success on 4; failure on 1 | 1 | 5% | 5 | 15% | 15 | 15% | 13 | 12% |
| Success on 3; failure on 2 | 3 | 16% | 1 | 3% | 11 | 11% | 13 | 12% |
| Success on 2; failure on 3 | 5 | 26% | 15 | 45% | 22 | 23% | 26 | 23% |
| Success on 1, failure on 4 | 6 | 32% | 6 | 18% | 15 | 15% | 19 | 17% |
| Success on 0, failure on 5 | 2 | 11% | 4 | 12% | 21 | 22% | 21 | 19% |
| Total | 19 | | 33 | | 97 | | 112 | |
| | | | | | | | | |
| **ORIGINAL NULL RESULTS** | | | | | | | | |
| Successful replication on all five criteria | 5 | 45% | 7 | 58% | 7 | 47% | 6 | 30% |
| Success on 4; failure on 1 | 2 | 18% | 1 | 8% | 2 | 13% | 2 | 10% |
| Success on 3; failure on 2 | 2 | 18% | 2 | 17% | 3 | 20% | 5 | 25% |
| Success on 2; failure on 3 | 2 | 18% | 2 | 17% | 2 | 13% | 2 | 10% |
| Success on 1; failure on 4 | 0 | 0% | 0 | 0% | 0 | 0% | 3 | 15% |
| Success on 0; failure on 5 | 0 | 0% | 0 | 0% | 1 | 7% | 2 | 10% |
| Total | 11 | | 12 | | 15 | | 20 | |

Five of the criteria we used to assess replications could be used for both positive results and null results. The number of papers, experiments, effects, and outcomes where replications were successful on various numbers of these criteria are shown for positive results (top) and null results (bottom). The five criteria were: (i) direction and statistical significance (p < 0.05); (ii) original effect size in replication 95% confidence interval; (iii) replication effect size in original 95% confidence interval; (iv) replication effect size in original 95% prediction interval; (v) meta-analysis combining original and replication effect sizes is statistically significant (p < 0.05). The data in this table are based on standardized mean difference (SMD) effect sizes. Very similar results are obtained when alternative strategies are used to aggregate the data (see Tables S8–S10 in **Supplementary file 1**).

has the virtue of ostensibly removing biasing influences of self-interest (pro or con) on the outcomes of the experiments. A skeptic, however, might suggest that original authors are essential for conducting the experiments because of particular skills or tacit knowledge they have for conducting the experiments. Indeed, this was part of a critique of the Reproducibility Project: Psychology (*Gilbert et al., 2016*; *Open Science Collaboration, 2015*). In that case, follow-up investigations did not support the claim that replication failures were due to deficiencies in replication quality. For example, Ebersole et al. repeated the potentially flawed replication protocols and developed revised protocols that were peer reviewed in advance by domain experts and original authors; they found that the replicability of original findings did not improve with the revised vs. original replication protocols (*Ebersole et al., 2020*; see also *Anderson et al., 2016*; *Nosek and Gilbert, 2017*). Nevertheless, the possibility of flaws in research is always present.

For the present replications, we attempted to minimize the likelihood of replication errors by using original materials whenever possible, employing large sample sizes, engaging expert peer review of the methodology in advance, and by preregistering the experiment design and analysis plan. Despite all this, we cannot rule out the possibility of methodological error in the replications. To facilitate further review, critique, and follow-up investigation, all replications in this meta-analysis are reported transparently with digital materials, data, and code openly available on OSF.

A failure to replicate could also occur even when both original and replication findings are 'correct'. Experimental contexts inevitably differ by innumerable factors such as the samples used, reagent and instrument suppliers, climate, software version, time of year, and physical environment. An experiment

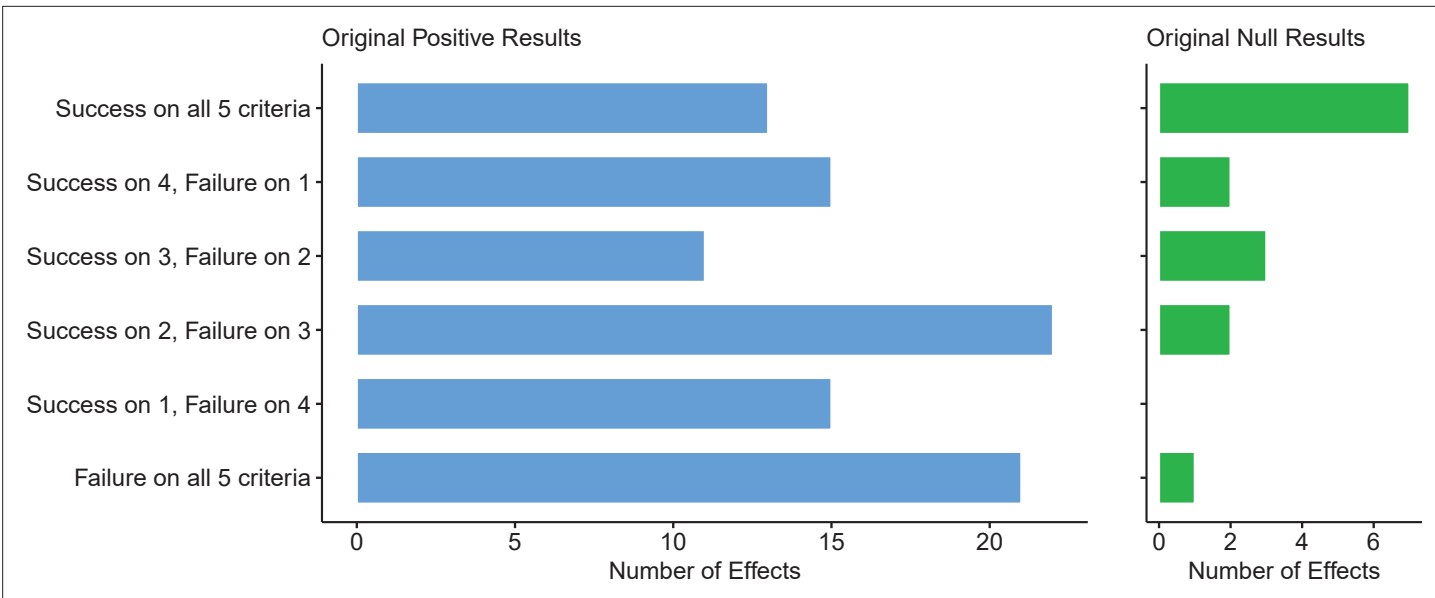

**Figure 6.** Assessing replications of positive and null effects across five criteria. Five of the criteria we used to assess replications could be used for both positive (n = 97) and null effects (n = 15). The number of effects where the replication was successful on all five criteria is shown by the top bar of each panel, with the second bar showing the number of effects where the replications were successful on four criteria, and so on: positive effects are shown in the left panel (blue bars), and null effects are shown in the right panel (green bars). The five criteria were: (i) direction and statistical significance (p < 0.05); (ii) original effect size in replication 95% confidence interval; (iii) replication effect size in original 95% confidence interval; (iv) replication effect size in original 95% prediction interval; (v) meta-analysis combining original and replication effect sizes is statistically significant (p < 0.05). Standardized mean difference (SMD) effect sizes are reported.

is a replication if the many differences between original and new experimental context are theoretically presumed to be irrelevant for observing evidence for the finding (*Nosek and Errington, 2020a*). The replication experiments underwent peer review in advance to arrive at a precommitment that they were good faith tests based on present understanding of the phenomena and the conditions necessary to observe evidence supporting them (*Nosek and Errington, 2020b*). However, differences that were deemed inconsequential during a priori peer review may be more critical than presently understood. After the replication results were known, some reviewers and commentators offered hypotheses for why the findings might have differed from the original (*Errington et al., 2021b*). This generative hypothesizing can be productive if it spurs additional investigations to test the new hypotheses. It can also be counterproductive if it is just rationalizing to preserve confidence in the original findings. *Hypothesizing ideas to test* is easily conflated with *providing evidence to explain*. Without follow-up investigation to test the hypotheses, that mix-up can promote overconfidence in original findings.

## What does a successful replication mean?

Successfully replicating a finding also does not render a verdict on its credibility. Successful replication increases confidence that the finding is repeatable, but it is mute to its meaning and validity. For example, if the finding is a result of unrecognized confounding influences or invalid measures, then the interpretation may be wrong even if it is easily replicated. Also, the interpretation of a finding may be much more general than is justified by the evidence. The particular experimental paradigm may elicit highly replicable findings, but also apply only to very specific circumstances that are much more circumscribed than the interpretation.

These possibilities are ordinary and unavoidable. Science makes progress by systematically producing and evaluating claims. Sometimes this leads to discoveries with broad generalizability and impact. Sometimes this leads to an understanding that is much more limited than the initial discovery. And, sometimes this leads to abandoning the effort because of persistent non-replicability or illumination of invalidity. Research produces its share of exhilaration with new discoveries and disappointments

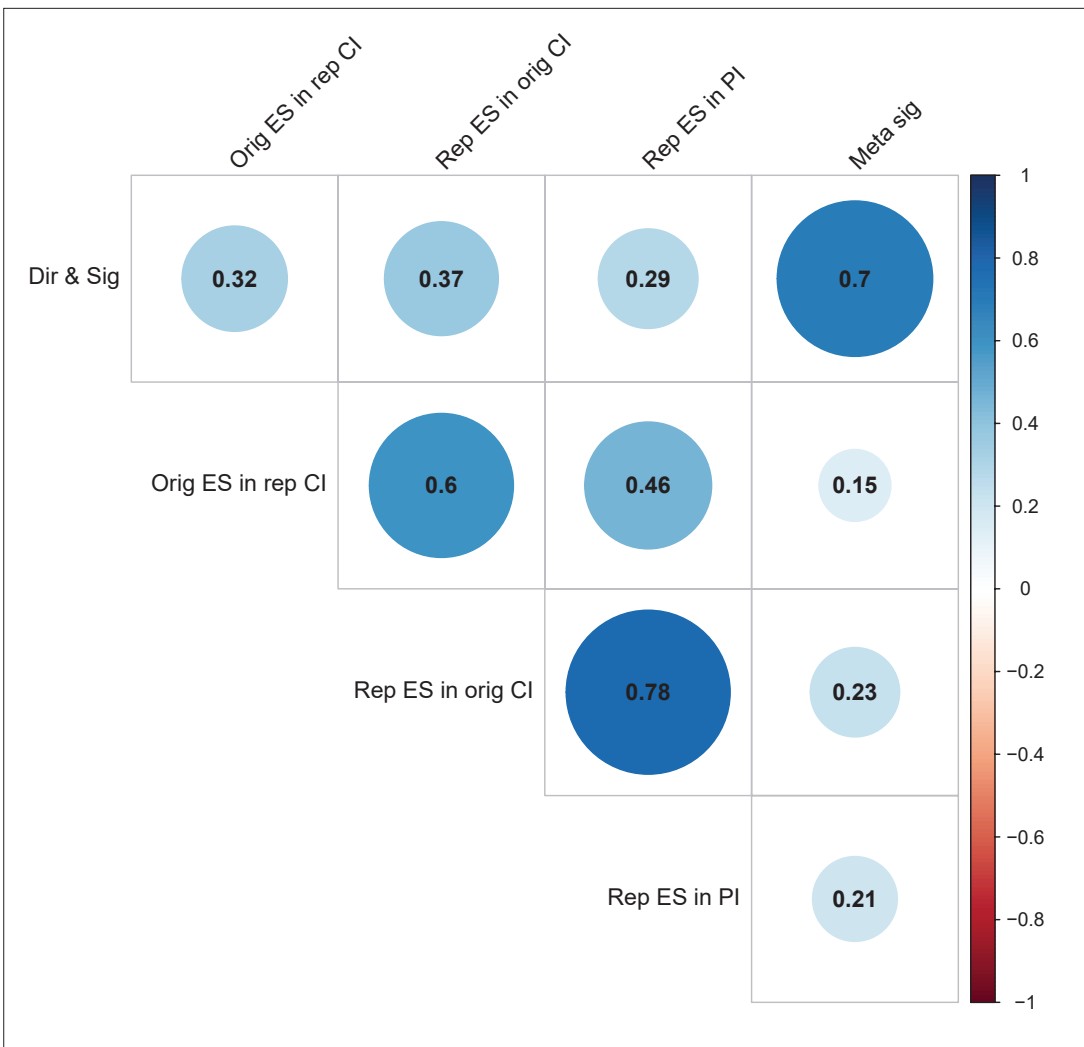

**Figure 7.** Correlations between five criteria for replication success. Point-biserial correlations among five criteria for evaluating replication success for the 112 original-replication pairs that could be evaluated on all five criteria: (i) same direction and statistical significance (Dir & Sig); (ii) original effect size in replication 95% confidence interval (Orig ES in rep CI); (iii) replication effect size in original 95% confidence interval (Rep ES in orig CI); (iv) replication effect size in 95% prediction interval (Rep ES in PI); (v) meta-analysis combining original and replication effect sizes gives significant effect (p < 0.05) (Meta sig). Correlations are color-coded (blue = positive; red = negative; see color bar), with the size of the circle being proportional to the magnitude of the correlation. The five criteria were all positively correlated with one another.

as some of them fade, but it is the continuous march away from ignorance that gets many scientists up each day excited to see which ideas will flourish.

## What replicates and what does not?

If we had better insights into the causes of replicability, or at least into the factors that correlate with replicability, we might be able to develop interventions that improve replicability. We explored five candidate moderators of replication success and did not find strong evidence to indicate that any of them account for variation in replication rates we observed in our sample. The clearest indicator of replication success was that smaller effects were less likely to replicate than larger effects, and this was particularly notable for animal experiments because they tended to have smaller original effect sizes than did non-animal experiments. Research into replicability in other disciplines has also found that findings with stronger initial evidence (such as larger effect sizes and/or smaller p-values) is more likely to replicate (*Nosek et al., 2021*; *Open Science Collaboration, 2015*), and it may be worth

exploring if other findings from other disciplines – such as more surprising findings being less likely to replicate – can be generalized to cancer biology. There are also unique qualities of research in cancer biology that could be related to replicability, and a number of ongoing projects exploring replication in preclinical research (*Amaral et al., 2019*; *Drude et al., 2021*) will add to the data presented here and increase our understanding of replication and translational success (*Chakroborty et al., 2020*). To facilitate such investigation, we have made all of the data for our meta-analysis available for reanalysis. Exploratory analyses with the dataset can help generate hypotheses about correlates of replicability that could be subjected to additional investigation.

## What have we learned about these findings?

After conducting dozens of replications, we can declare definitive understanding of precisely zero of the original findings. That may seem a dispiriting conclusion from such an intense effort, but it is the reality of doing research. Original findings provided initial evidence, replications provide additional evidence. Sometimes the replications increased confidence in the original findings, sometimes they decreased confidence. In all cases, we now have more information than we had. In no cases, do we have all the information that we need. Science makes progress by progressively identifying error and reducing uncertainty. Replication actively confronts current understanding, sometimes with affirmation, other times signaling caution and a call to investigate further. In science, that's progress.

## What have we learned about replicability of preclinical cancer biology?

We adopted a wide and shallow approach with mostly single replications of many findings. We learned a little about each finding and more about replicability of preclinical cancer biology in general. If we had conducted a narrow and deep approach with the same resources, many replications of few findings, we would have learned more about the individual findings and less about cancer biology in general. The present study provides substantial evidence about the replicability of findings in a sample of high-impact papers published in the field of cancer biology in 2010, 2011, and 2012. The evidence suggests that replicability is lower than one might expect of the published literature. Causes of non-replicability could be due to factors in conducting and reporting the original research, conducting the replication experiments, or the complexity of the phenomena being studied. The present evidence cannot parse between these possibilities for any particular finding. But, there is substantial evidence of how the present research culture creates and maintains dysfunctional incentives and practices that can reduce research credibility in general. There are also reforms emerging that could address those challenges and potentially improve replicability.

It is unlikely that the challenges for replicability have a single cause or a single solution. Selective reporting, questionable research practices, and low-powered research all contribute to the unreliability of findings in a range of disciplines (*Button et al., 2013*; *Franco et al., 2014*; *Ioannidis, 2005*; *John et al., 2012*). These challenges may be compounded by researchers not receiving sufficient training in statistical inference and research methodology (*Van Calster et al., 2021*). Moreover, as reported in the companion paper (*Errington et al., 2021b*), failures to document research protocols and to share data, materials (digital and physical), and code are hindering efforts to increase replicability (see also *Lemmon et al., 2014*; *Serghiou et al., 2021*; *Stodden et al., 2018*; *Vines et al., 2014*). These issues are exacerbated by dysfunctional incentives in the research culture that favor positive, novel, tidy results, even at the expense of rigor, accuracy, and transparency. In a system that rewards researchers with publications, grants, and employment for producing exciting and innovative results, it is no surprise that the literature is filled with exciting and innovative results. A system that also rewarded rigor, accuracy, and transparency as counterweights might stem some of the most unproductive impacts on research credibility, improve the culture, and accelerate progress.

There are solutions available that could have a substantial positive impact on improving research practices. Technologies provide mechanisms for research planning, preregistration, and sharing data, materials, and code (*Baker, 2019*; *Cragin et al., 2010*; *Heinl et al., 2020*; *Horai et al., 2010*; *Lindsay et al., 2016*; *Soderberg, 2018*). Training services can assist researchers in maximizing the value of these technologies and advance their understanding of research methodology and statistical inference (*Casadevall et al., 2016*; *Teal et al., 2015*; *Wilson, 2014*). Incentive focused innovations such as Registered Reports can shift reward away from achieving exciting results and toward asking important questions and designing rigorous studies to investigate them (*Chambers, 2019*; *Scheel et al., 2020*;

*Soderberg et al., 2020*). Signals such as badges to acknowledge open practices can increase visibility of these behaviors to facilitate changing norms within research communities (*Kidwell et al., 2016*). Finally, journals, funders, and institutions can assess the research cultures and incentives that they create and maintain and introduce new policies that are values-aligned to promote research rigor and transparency (*Hatch and Curry, 2020*; *Nosek et al., 2015*). Improvements to infrastructure, training, norms, incentives, and policies are each necessary, and none individually sufficient, to foster a research culture that rewards rigor, transparency, and sharing, to ultimately reduce friction and accelerate progress.

## Limitations

The present investigation provides evidence about the challenges involved in replicating a sample of experiments from high-impact papers in cancer biology. This could have implications for research on cancer biology and preclinical life sciences research more generally. However, there are important cautions about the selection and replication process that make the generalizability of these findings unclear. The systematic selection process identified 193 experiments from 53 high-impact papers published in 2010, 2011, and 2012. We experienced a variety of barriers to preparing and conducting the replications (*Errington et al., 2021b*). We do not know if those barriers produced a selection bias that altered our likelihood of successful replication, for better or worse. Moreover, replications were conducted on a subset of experiments from the papers. We cannot rule out the possibility that we inadvertently selected experiments from within the papers that were less likely to replicate.

One protection against the possibility of selection bias was the fact that all the experimental designs and protocols for the replications were peer reviewed by experts in advance, which sometimes resulted in changes to the experiments selected for replication. There was no systematic observation of biased selection of experiments that were more or less likely to replicate. More generally, selecting high-impact papers may result in biasing influence: for example, if papers reporting findings that have lower prior odds of being correct are more likely to gain attention and citations, then selecting these papers may overestimate failures to replicate for findings in general. Alternatively, if high-impact papers, most of which are published in 'prestigious' journals, are more likely to receive and withstand scrutiny prior to publication, then selecting these papers may underestimate failures to replicate for findings in general. In any case, compared to the baseline presumption that the published literature is credible and replicable, our findings from a systematically selected sample of the literature suggest that there is room for improvement.

## Conclusion

No single effect, experiment, or paper provides definitive evidence about its claims. Innovation identifies possibilities. Verification interrogates credibility. Progress depends on both. Innovation without verification is likely to accumulate incredible results at the expense of credible ones and create friction in the creation of knowledge, solutions, and treatments. Replication is important for research progress because it helps to separate what we know from what we think we know.

The surprisingly high rate of failures to replicate in this study might be an indicator of friction in the research process inhibiting the successful translation of basic and preclinical research. For example, promising animal experiments are often the basis of launching clinical trials. The low replication success rates and small effect sizes we found reinforce prior calls for improving the rigor and transparency of preclinical research to improve the allocation of limited resources and to protect human participants from being enrolled in trials that are based on weak evidence (*Landis et al., 2012*; *Perrin, 2014*; *Steward et al., 2012*).

Stakeholders from across the research community have been raising concerns and generating evidence about dysfunctional incentives and research practices that could slow the pace of discovery. This paper is just one contribution to the community's self-critical examination of its own practices. Science pursuing and exposing its own flaws is just science being science. Science is trustworthy because it does not trust itself. Science earns that trustworthiness through publicly, transparently, and continuously seeking out and eradicating error in its own culture, methods, and findings. Increasing awareness and evidence of the deleterious effects of reward structures and research practices will spur one of science's greatest strengths, self-correction.

## Materials and methods
### Paper, experiment, and effect selection strategy

Of the 193 experiments from 53 papers selected for replication, a total of 50 experiments from 23 papers were completed with 158 unique effects. See *Errington et al., 2014* and *Errington et al., 2021b* for full descriptions of the sampling strategy, methodology, and challenges in conducting the replications. Briefly, we asked the original authors for data, materials, and advice on the protocols to maximize the quality of the replications. If we were able to design complete replication protocols, identify labs to conduct the experiments, and obtain materials and reagents to perform the experiments, we then submitted the protocols for peer review at *eLife* as Registered Reports, a publishing format in which peer review is conducted prior to observing the experimental outcomes to maximize quality of methodology, ensure precommitment and transparency of registered methods and analysis plans, and to minimize publication bias and selective reporting (*Chambers, 2019*; *Nosek and Lakens, 2014*). Thus, all selected experiments and effects of interest, protocol details, material choices, and analytical strategies were peer reviewed in advance. In total, we published 29 Registered Reports including 87 planned experiments. For each experiment, effects of interest were identified with sample sizes determined a priori for at least 80% power to detect the original effect size. For 18 of the original papers we were able to implement modifications to complete the proposed experiments resulting in a total of 17 published Replication Studies with the one rejected Replication Study posted as a preprint (*Pelech et al., 2021*). For the five original papers where we completed some of the experiments the results of the completed replications were reported in an aggregate paper (*Errington et al., 2021a*). In total there were 158 effects with replication outcomes from 50 experiments (effects per experiment: M = 3.2; SD = 2.4; range = 1–13) that came from 23 papers (effects per paper: M = 3.9; SD = 2.8; range = 1–13). Some of the 158 effects had internal replications (n = 19) in which we conducted multiple replications (range = 2–3) of the same original experiment leading to 188 total outcomes considering the internal replications separately. All of these outcomes are available on OSF (https://osf.io/e5nvr/). That database is the basis of the meta-analysis reported in this paper. All replication protocols, materials, data, and outcomes are documented, archived, and publicly accessible to maximize transparency, accountability, and reproducibility of this project and are available at: https://osf.io/collections/rpcb/. All individual papers published as part of this project are available at *eLife* (https://elifesciences.org/collections/9b1e83d1/reproducibility-project-cancer-biology).

### Calculation and extraction of statistical data

Papers and experiments were coded as described in *Errington et al., 2021b*. All original outcome data were calculated using either the shared original raw data, shared original summary data, extracted summary data from original papers, or original statistical variables from original papers. Summary data from all original experiments were reported in the associated Registered Reports and used for power calculations. Replication outcome data were calculated using the replication raw data with the outcomes and summary data reported in the associated Replication Studies. There was variation in reporting across outcomes that constrained what kinds of comparisons could be made between original and replication findings (e.g., 117 [74%] of the 158 original effects reported numerical results [e.g., graphs of quantified data or statistical tests] while other effects may have been reported as a representative image without any information about variability or an associated statistical inference). And of the 158 original effects, 86% were positive (i.e., interpreted as observing a relationship or impact of an intervention) and 14% were null (i.e., interpreted as not observing a meaningful relationship or impact of an intervention). For each outcome the statistical tests, when possible, were calculated based on the native structure of the original or replication data with a common shared effect size calculated for each original and replication pair. In 19 cases (5 where the original was a numerical result and 14 where the original was representative), this was not possible for the replication, meaning only the 'same direction' criterion was able to be determined. There was also one case where the original point estimate from the representative image in the original study was not able to be determined, meaning only the 'same direction' and 'significance agreement' criteria could be assessed, but not others (e.g., if the original image estimate was within the 95% confidence interval of the replication). For the cases where there were statistical tests and effect sizes, they ranged from types of standardized mean differences (SMDs) (e.g., Cohen's d) and non-parametric equivalents (e.g., Cliff's delta) across the range of outcomes. To facilitate effect-size conversions to approximate the SMDs scale we calculated,

or approximately converted, all effect sizes to the SMD scale and recalculated statistical tests where necessary. This allows for meta-analysis and aggregation across a wider range of outcomes, although has the risk of distorting the results. As such, we conducted analyses below using the outcomes in the native or SMD scale, which gave similar patterns (see Tables S1–S3, S4–S6, S8–S10 in *Supplementary file 1*). Data dictionaries describing all of the variables are available at https://osf.io/e5nvr/.

### Effect size conversions

For numerical results, we extracted effect sizes on the following SMD scales for all outcomes: Cohen's d (110 outcomes), Cohen's dz (15 outcomes), Glass' delta (24 outcomes), hazard ratio ( outcomes), and Pearson's r (6 outcomes). The first three effect size scales are types of SMDs, although their interpretations are somewhat different from one another. We approximately converted hazard ratios (*Hasselblad and Hedges, 1995*) and Pearson's correlations (*Mathur and VanderWeele, 2020a*) to the SMD scale. Converting Pearson's correlations calculated with continuous independent variables to SMDs requires specifying the size of contrast in the independent variable that is to be considered. We selected a contrast size of 1 standard deviation on the independent variable throughout.

### Accounting for nested data

The data had a hierarchical structure, with effects nested within experiments nested with papers. We first calculated five pairwise metrics of replication success (detailed below) at the effect level, which was the finest-grained level of analysis. To account for this nesting structure, we first used fixed-effects meta-analysis to combine internal replications within a given paper, experiment, and effect, reflecting the assumption that they were testing the same effect size. We report aggregated findings at each level of analysis for ease of comprehension and multi-level analysis that account for the nested structure. Effects were meta-analytically combined into experiments with random-effect models reflecting the fact that the effects could be heterogeneous within an experiment. And experiments were meta-analytically combined into papers with random-effect models for the same reason. Original positive and null effects were kept separate in aggregating into experiments and papers. This was done for effects on the SMD scale (main paper results presented this way) and for effects on the native scale (where possible; Tables S1, S4, S8 in *Supplementary file 1*). We also summarized these effect-level metrics at the experiment and paper level by calculating percentages to summarize binary metrics of replication success, and by using harmonic mean p-values to summarize the continuous metric (*Wilson, 2019*). This was done for effects on the native scale (Tables S2, S5 and S9 in *Supplementary file 1*) and the SMD scale (Tables S3, S6, and S10 in *Supplementary file 1*). When conducting meta-regression analyses across pairs (i.e., when effect sizes could be determined for both original and replication effects), we accounted for the hierarchical structure using robust inference (*Bell and McCaffrey, 2002*; *Pustejovsky and Tipton, 2017*) to account for the nesting structure as described in 'Assessing candidate moderators of replication success metrics' below.

### Pairwise metrics of replication success

We evaluated replication success with a variety of outcomes that have been employed in prior replication studies: (1) We evaluated whether the sign of the replication estimate agreed with that of the original effect ('same direction'). This rudimentary metric does not account for effect sizes nor statistical precision, but was useful because we could compute it for even the non-quantitative pairs such as when original experiments reported only representative images. (2) We assessed whether the replication had a p-value less than 0.05 and an estimate in the same direction as the original ('significance agreement'). (3 and 4) We assessed whether the original and replication estimates were inside the 95% confidence intervals of the other. (5) We assessed whether the replication estimate was inside the 95% prediction interval of the original, and formally assessed the degree of statistical inconsistency between the replication and the original using the metric $p_{orig}$, which can be viewed as a p-value for the hypothesis that the original and the replication had the same population effect size (*Mathur and VanderWeele, 2020b*). (6) We estimated the difference in estimates between the replication and the original after transforming all effect sizes to a comparable scale. Finally, regarding evidence strength for the effects under investigation, (7) we calculated a pooled estimate from combining the replication and original via fixed-effects meta-analysis. This is equivalent to pooling individual observations from the replication and the original data and can be viewed as an updated estimate of the effect size if

the original experiment and the replication experiment are treated as equally informative. We chose a fixed-effect model rather than a random-effects model primarily for consistency with our assumption throughout all main analyses that there was no within-pair heterogeneity. Also, a random-effects model typically cannot adequately estimate heterogeneity with only two effects (*Langan et al., 2019*). Additionally, in the specific context of pooling a single replication effect with a single original effect, we view the natural target of statistical inference as the mean effect size for those two effects rather than for a hypothetical larger population of effects on the same topic from which the original and replication were drawn. For this purpose, inference from a fixed-effect model is appropriate (*Rice et al., 2017*). In any case, reanalysis with a random-effects model has modest impact on the estimates and does not alter the substantive conclusions. Rationales, including the strengths and limitations of each of these metrics, are presented in context of the outcomes in the Results section.

We conducted all statistical analyses with R software (RRID:SCR_001905), version 4.0.3 (*R Development Core Team, 2021*). All statistical analyses were determined *post hoc*. We also used metafor software (RRID:SCR_003450), version 2.4–0 to conduct the meta-analyses.

## Assessing candidate moderators of replication success metrics

We assessed whether five candidate moderators were associated with each indicator of replication success. The moderators were: (1) animal experiments vs. non-animal (i.e., in vitro) experiments; (2) whether at least one of the replication labs was a contract research organization; (3) whether at least one of the replication labs was an academic research core facility; (4) whether the original authors shared materials with the replicating labs (coded as 'no', 'yes', or 'not requested', with the latter indicating that we did not need to request materials); and (5) the quality of methodological clarifications made by the original authors upon request from the replicating labs (coded as 0 = 'no response or not helpful' to 5 = 'extremely helpful' and analyzed as a continuous variable). We intended to include a sixth moderator, which described the extent to which we were successful at implementing any needed protocol changes (analyzed as a categorical variable with categories 'changes were moderately, mostly, or completely implemented', 'changes were less than moderately implemented', and 'no changes needed'). However, all but one outcome were rated in the first category, making the model inestimable with this variable included. We therefore excluded this moderator from the model.

To assess whether the five moderators were associated with these metrics of replication success across quantitative pairs, we used multi-level models to regress each pairwise metric on all candidate moderators simultaneously, accounting for clustering within experiments and papers. Some pairwise metrics had variances associated with them (i.e., the difference in estimate and the pooled estimate), whereas the others did not. For the metrics that did not have variances, we obtained point estimates from a standard multi-level model containing random intercepts of experiments nested within papers. For the metrics that did have variances, we obtained point estimates from an equivalent model that also weighted pairs by the inverse variance of the outcome variable (i.e., the replication success metric); this model is a form of random-effects meta-analysis. In both cases, we used CR2 robust standard errors in order to relax the distributional assumptions of parametric mixed models.

These models estimated the average difference in each metric of replication success that was associated with each candidate moderator, while holding constant all the other moderators. We report inference both with and without Bonferroni corrections for multiplicity. The Bonferroni corrections adjusted for multiplicity across the five moderator coefficients per metric of replication success, but did not adjust for the multiple metrics of replication success because these metrics were of course highly correlated with one another, and were sometimes arithmetically related to one another.

## Sensitivity analysis

All of the replication success metrics assume that there is no heterogeneity within quantitative pairs. This is a limitation because there could be substantive differences, including unmeasured moderators, between a given original effect and its replication that could produce genuine substantive differences between the two estimates. Such heterogeneity is not uncommon, even if it is relatively small, in multisite replications in which heterogeneity can be directly estimated (*Ebersole et al., 2020*; *Klein et al., 2018*). As such, the metrics will typically underestimate replication success when there is heterogeneity within pairs (*Mathur and VanderWeele, 2020b*). As a sensitivity analysis, we also evaluated evidence for inconsistency by calculating $p_{orig}$, constructing prediction intervals, and estimating

the expected significance agreement across pairs under the assumption that there was within-pair heterogeneity with standard deviation 0.21 on the SMDs scale, an estimate we obtained from reanalyzing a review of multisite replications (*Mathur and VanderWeele, 2020b*; *Olsson-Collentine et al., 2020*). This sensitivity analysis yielded similar results and conclusions to the main analyses for these three metrics of replication success (see Tables S7 and S11 in *Supplementary file 1*), likely because the estimated heterogeneity was small relative to the original and replication standard errors. Future research could attempt to directly estimate heterogeneity in these research contexts.

## Note

All *eLife* content related to the Reproducibility Project: Cancer Biology is available at: https://elife-sciences.org/collections/9b1e83d1/reproducibility-project-cancer-biology.

All underlying data, code, and digital materials for the project is available at: https://osf.io/collections/rpcb/.

## Acknowledgements

We thank Fraser Tan, Joelle Lomax, Rachel Tsui, and Stephen Williams for helping in coordination efforts during the course of the project. We thank all Science Exchange providers who provided their services and all employees at Science Exchange and the Center for Open Science who contributed to administrative and platform development efforts that enabled this project to occur.

## Additional information

### Competing interests

Timothy M Errington, Alexandria Denis: Employed by the Center for Open Science, a non-profit organization that has a mission to increase openness, integrity, and reproducibility of research. Courtney K Soderberg: Was employed by the Center for Open Science, a non-profit organization that has a mission to increase openness, integrity, and reproducibility of research. Nicole Perfito: Was employed by and holds shares in Science Exchange Inc. Elizabeth Iorns: Employed by and holds shares in Science Exchange Inc. Brian A Nosek: Employed by the nonprofit Center for Open Science that has a mission to increase openness, integrity, and reproducibility of research. The other author declares that no competing interests exist.

### Funding

| Funder | Grant reference number | Author |
|---|---|---|
| Arnold Ventures | | Timothy M Errington Brian A Nosek |

The funders had no role in study design, data collection and interpretation, or the decision to submit the work for publication.

### Author contributions

Timothy M Errington, Conceptualization, Data curation, Formal analysis, Investigation, Methodology, Project administration, Supervision, Validation, Visualization, Writing – original draft, Writing – review and editing; Maya Mathur, Formal analysis, Methodology, Validation, Visualization, Writing – original draft, Writing – review and editing; Courtney K Soderberg, Methodology, Visualization, Writing – review and editing; Alexandria Denis, Data curation, Investigation; Nicole Perfito, Conceptualization, Project administration, Writing – review and editing; Elizabeth Iorns, Conceptualization, Funding acquisition, Project administration, Writing – review and editing; Brian A Nosek, Conceptualization, Funding acquisition, Methodology, Project administration, Supervision, Visualization, Writing – original draft, Writing – review and editing

### Author ORCIDs

Timothy M Errington  http://orcid.org/0000-0002-4959-5143
Maya Mathur  http://orcid.org/0000-0001-6698-2607

Elizabeth Iorns http://orcid.org/0000-0002-5515-1258
Brian A Nosek http://orcid.org/0000-0001-6797-5476

**Decision letter and Author response**
Decision letter https://doi.org/10.7554/eLife.71601.sa1
Author response https://doi.org/10.7554/eLife.71601.sa2

## Additional files

### Supplementary files
• Supplementary file 1. Tables S1–S11.

• Transparent reporting form

### Data availability
All experimental details (e.g., additional protocol details, data, analysis files) of the individual replications and data, code, and materials for the overall project are openly available at https://osf.io/collections/rpcb/. Master data files, containing the aggregate coded variables, are available for exploratory analysis at https://osf.io/e5nvr/.

The following dataset was generated:

| Author(s) | Year | Dataset title | Dataset URL | Database and Identifier |
|---|---|---|---|---|
| Errington TM, Denis A | 2021 | Replication Data from the Reproducibility Project: Cancer Biology | https://doi.org/10.17605/osf.io/e5nvr | Open Science Framework, 10.17605/osf.io/e5nvr |

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
