## [Decision Letter]

**Decision letter after peer review:**

Thank you for submitting your article "Investigating the Replicability of Cancer Biology" to *eLife*. Your article has been reviewed by three peer reviewers, and the evaluation has been overseen by an *eLife* Reviewing Editor (Renata Pasqualini), an *eLife* Senior Editor (Eduardo Franco), and the *eLife* Features Editor (Peter Rodgers). The following individuals involved in review of your submission have agreed to reveal their identity: Paul Glasziou (Reviewer #1); Malcolm MacLeod (Reviewer #2); Tudor Oprea (Reviewer #3).

The reviewers and editors have discussed the reviews and we have drafted this decision letter to help you prepare a revised submission.

Summary

This is an excellent summary of a landmark set of replication research studies which builds on the large-scale replication studies done in psychology and extends them to cancer biology. The authors have replicated 50 important experiments published in high impact Cancer biology papers and have done so in both a rigorous and transparent way. These important findings should trigger a rethink in how we do cancer biology research and indeed all research. Of particular note is their finding that "original null results were twice as likely to mostly replicate successfully (80%) as original positive results (40%)." However, there are a number of points that need to be addressed to make the article suitable for publication. The presentation of the results could also be improved.

Essential revisions

1. The violin plots (eg Figure 1) do not indicate the number of experiments contributing to each panel. Eg for Original null papers, I think n=11 … yet the violin plot implies greater precision in the shape of the distribution. Is there a way to fix this? And are quantiles best described as "25","50" etc or Q1, Q2 …

2. The choice of ES measure will introduce error when n is small. You say that you re-do the analysis with native data, but in the supplementary table the n for native data appears to be the same as for the variance based measure. If so, why not just use the native data? If not, can you clarify? I may have missed something …

3. For the ~30 animal studies which were included, could you report the proportion that were randomised, blinded etc, and report whether this was associated with differences in observed replication rates? I was going to do this myself, but thought you'd rather have the review sooner!

4. Figure 2: The distribution curves seem wrong – it appears (to my eye) that these include some points from the full dataset which are not included in the panel (because of cropping axes), and this distracted me. For instance, replication effect size p>0.05 there are two "bumps" above the central spike, only one of which appears to relate to a point shown on the graph.

5. Interquartile ranges are offered as a number rather than a range … and if the distribution is not symmetrical, this is losing information.

6. Line 59 – some would quibble that knowing that lab does indeed change your confidence in the reported findings. That is, a lab with a track record of getting things wrong should not start from the same position of credibility as the obverse.

7. Line 62 – can you define validity?

8. Line 67. "successive replications …" a faithful replication will presumably share the same confounding influences, if these are part of the design (ie features not bugs). But I agree successive replications do reduce the possibility that the findings are due to some latent confounding variable.

9. Line 70 et seq: I liked this, but I wonder if the component "misunderstanding or underdeveloped theory" will land with life scientists unless more richly described.

10. Line 179: I think this depends on your scale – if it is "detected" or "not detected" and you have very low sensitivity, most will be nil. Of course, the chances that the underlying biology does not change one iota is close to zero.

11. Measures of replication – is the most relevant not the question of whether the ES(rep) lies within the 95% prediction interval of the first experiment? I know that this "advantages" imprecise originator experiments, but we take that into account when we use the information therein. This is somewhat related to Hanno Wuerbul's claim that heterogenous experiments are better, because they are better predictors of what will happen in a further experiment – largely, but not exclusively, because their prediction intervals tend to be wider.

12. The authors write: " After conducting dozens of replications, we can declare a definitive understanding of precisely zero of the original findings."

– In my opinion, there should have been more consequences. For example, those papers that failed the reproducibility test. Shouldn't there be retraction notices? Or at least, "expressions of concern" to the Publishers? If there is no real consequence, the only thing we learn is that there's monkey business in science. [This point is taken from the report of Tudor Oprea: please see below for the full report].

Please comment on what you think authors and journals should do when a paper they have published fails to replicate.

Points related to presentation

A. Figures 1, 2, and 3 are the P-value densities for the original and replication experiments which is extremely informative, but I would prefer the axis for the p-values to the more conventional x-axis. This may be a personal preference, but I found rotating the figure by 90 degrees made it much easier to read – please consider making this change.

B. It is not clear that figure 2B adds anything to the article – please consider deleting this panel.

C. Figure 5 (and similar figure supplements) should use the same scaling on both axes so that the diagonal "equality" line is at 45 degrees.

D. Tables 1, 2, and 3 set out the results in great detail, but are very dense with numbers. I wondered if these might be better as a hybrid figure/table (which can include the raw numbers inside the bars of the boxes or histograms). For example, see – https://stackoverflow.com/questions/6644997/showing-data-values-on-stacked-bar-chart-in-ggplot2.

---

## [Author Response]

Essential revisions1. The violin plots (eg Figure 1) do not indicate the number of experiments contributing to each panel. Eg for Original null papers, I think n=11 … yet the violin plot implies greater precision in the shape of the distribution. Is there a way to fix this? And are quantiles best described as "25","50" etc or Q1, Q2 …

Fixed.

2. The choice of ES measure will introduce error when n is small. You say that you re-do the analysis with native data, but in the supplementary table the n for native data appears to be the same as for the variance based measure. If so, why not just use the native data? If not, can you clarify? I may have missed something …

We look at the data two ways: one where we keep the native effect size type for each individual effect and one where we convert effect size types to a standardized mean difference (SMD) scale. The sample size n does not change during these, rather we are converting effect sizes to a SMD scale (e.g., hazard ratios to an SMD scale as described in Hasselblad and Hedges, 1995). We report both as conversions can potentially distort the results; however, we find the outcome is similar regardless of which approach is used. The reason we report the SMD scale in the main results is to keep all tables, figures, etc comparable to each other since some tables (Table 3, 5, and 6) and figures (Figures 2, 3, and 5) include effect sizes that are reported relative to each other, which can only be done when effect sizes are on the SMD scale.

3. For the ~30 animal studies which were included, could you report the proportion that were randomised, blinded etc, and report whether this was associated with differences in observed replication rates? I was going to do this myself, but thought you'd rather have the review sooner!

The reviewer rightly points out that randomization, blinding, and sample size planning are important factors that have been identified as potentially improving replicability of findings. Unfortunately, in this case, there is not enough variation in the original findings to examine these as potential moderators. For the 36 animal effects across 15 experiments, none reported blinding, none reported determining sample size a priori, and one experiment (for two effects) reported randomization.

For completeness sake, blinding and randomization were not reported for all replication studies. For the 36 animal effects, 11 reported blinding (5 of 15 experiments), 28 reported randomization (13 of 15 experiments), and all 36 (15 of 15 experiments) reported determining sample size a priori. We added a brief description of these data at the end of the second paragraph of the section “Comparing animal versus non-animal experiments”.

4. Figure 2: The distribution curves seem wrong – it appears (to my eye) that these include some points from the full dataset which are not included in the panel (because of cropping axes), and this distracted me. For instance, replication effect size p>0.05 there are two "bumps" above the central spike, only one of which appears to relate to a point shown on the graph.

Thank you for catching this. These have been fixed so the distribution curves are only for the cropped data.

5. Interquartile ranges are offered as a number rather than a range … and if the distribution is not symmetrical, this is losing information.

Fixed.

6. Line 59 – some would quibble that knowing that lab does indeed change your confidence in the reported findings. That is, a lab with a track record of getting things wrong should not start from the same position of credibility as the obverse.

We agree that, in practice, researchers can gain a reputation for high or low credibility research. We refer here to scientific ideals and provide the qualifier “supposed to” for the quibble:

“Credibility of research claims are not supposed to be contingent on the reputation of their originator; credibility is supposed to be based on the quality of the evidence itself, including replicability.”

7. Line 62 – can you define validity?

Fixed.

8. Line 67. "successive replications …" a faithful replication will presumably share the same confounding influences, if these are part of the design (ie features not bugs). But I agree successive replications do reduce the possibility that the findings are due to some latent confounding variable.

Yes, we agree. We added “But successive replications may never eliminate some confounding influences or invalidity in design.” to the end of the paragraph.

9. Line 70 et seq: I liked this, but I wonder if the component "misunderstanding or underdeveloped theory" will land with life scientists unless more richly described.

We elaborated this statement to say: “It is possible that the methodology necessary to produce the evidence is not sufficiently defined or understood. This could mean that the original authors misunderstand what is necessary or sufficient to observe the finding, or that the theoretical explanation for why the finding occurred is incorrect.”

10. Line 179: I think this depends on your scale – if it is "detected" or "not detected" and you have very low sensitivity, most will be nil. Of course, the chances that the underlying biology does not change one iota is close to zero.

Yes, it is possible to have low sensitivity measures that detect no variation at all even if there might be some variation that is undetectable. This does not have direct implications for our use of “nil” here to refer to literally zero because we observed variation in measurements for virtually all outcome variables.

11. Measures of replication – is the most relevant not the question of whether the ES(rep) lies within the 95% prediction interval of the first experiment? I know that this "advantages" imprecise originator experiments, but we take that into account when we use the information therein. This is somewhat related to Hanno Wuerbul's claim that heterogenous experiments are better, because they are better predictors of what will happen in a further experiment – largely, but not exclusively, because their prediction intervals tend to be wider.

Patil et al., (2016) agree with this argument, but there is substantial disagreement among methodologists about the best metric for interpreting replication success. For example, with extremely high precision tests, it becomes very easy to fail to replicate using prediction intervals even when the effect is observed reliably and precisely and is just a bit smaller or larger than the original. For many research applications, a bit of variation in overall effect size would not be consequential for theoretical interpretation. In any case, our position is that there is no singular indicator of replication success--which is preferred is somewhat dependent on setting a liberal versus conservative criterion, and somewhat dependent on which specific question one is aiming to answer. As such, we present a variety of outcome criteria and avoid asserting that one is the best.

12. The authors write: " After conducting dozens of replications, we can declare a definitive understanding of precisely zero of the original findings."

In my opinion, there should have been more consequences. For example, those papers that failed the reproducibility test. Shouldn't there be retraction notices? Or at least, "expressions of concern" to the Publishers? If there is no real consequence, the only thing we learn is that there's monkey business in science. [This point is taken from the report of Tudor Oprea: please see below for the full report].

Please comment on what you think authors and journals should do when a paper they have published fails to replicate.

We don’t believe that a failure to replicate on its own requires any action by the journal. The primary reason we believe that is because of the selected quote. A single replication study is not (maybe we could agree to ‘rarely’) a definitive statement of the confidence we should or should not have in the original finding. As much as we attempted to conduct rigorous, high-powered, transparent replication studies, it is possible that something went wrong and the original study findings are replicable under other circumstances. So, our position is that a failure to replicate should prompt a closer look and, often, additional investigation to demonstrate replicability if researchers perceive possible improvements that could establish it.

Failures to replicate do increase our scepticism of the original findings, but if journals are perceived as definitive arbiters of truth - accepted if replicable, retracted if unreplicable - it would foster further dysfunction in the research culture because individual studies do not provide definitive evidence. A more functional approach by journals would be to embrace uncertainty of every paper and publish replication studies and accumulation of evidence to support resolving uncertainty rather than demand innovation in every publication.

More generally, we do appreciate Tudor Opera’s comments about the need to improve self-correction in the literature. However, we believe that the impact of this particular paper will be strongest with a focus on the general implication of unreplicability rather than focusing on the replicability of the particular papers and findings that happened to be sampled.

Points related to presentationA. Figures 1, 2, and 3 are the p-value densities for the original and replication experiments which is extremely informative, but I would prefer the axis for the p-values to the more conventional x-axis. This may be a personal preference, but I found rotating the figure by 90 degrees made it much easier to read – please consider making this change.

Figures 1 (p-value densities) and 3 (effect size densities) were flipped so the values are along the x-axis opposed to the y-axis.

B. It is not clear that figure 2B adds anything to the article – please consider deleting this panel.

Panel 2A shows most of the entire distribution of effect sizes. As a consequence, it is visually dominated by the most extreme original effect sizes. Figure 2B zooms in on where most of the original and replication studies effect sizes occurred so that the reader can perceive the pattern of decline more clearly where most of the data are. We find it useful for our own understanding of the findings and anticipate that some readers will likewise.

C. Figure 5 (and similar figure supplements) should use the same scaling on both axes so that the diagonal "equality" line is at 45 degrees.

Fixed.

D. Tables 1, 2, and 3 set out the results in great detail, but are very dense with numbers. I wondered if these might be better as a hybrid figure/table (which can include the raw numbers inside the bars of the boxes or histograms). For example, see – https://stackoverflow.com/questions/6644997/showing-data-values-on-stacked-bar-chart-in-ggplot2.

Thank you for this suggestion. We kept each table as is for full transparency and have included a new figure (Figure 7) which summarizes the main findings from Table 7 for effects.